# Influence of snowpack properties and local incidence angle on SAR signal depolarization: a mathematical model for high-resolution snow depth estimation

Alberto Mariani<sup>1,2</sup>, Jacopo Borsotti<sup>3</sup>, Franz Livio<sup>2</sup>, Giacomo Villa<sup>1</sup>, Martin Metzger<sup>1</sup>, and Fabiano Monti<sup>1</sup>

<sup>1</sup>Alpsolut S.r.l., Via Saroch 1098/a, 23041 Livigno, Italy

<sup>2</sup>University of Insubria, Department of Science and High Technology, Via Valleggio 11, 22100 Como, Italy

<sup>3</sup>University of Parma, Department of Mathematical, Physical and Computer Sciences, Parco Area delle Scienze 53/A, 43124 Parma, Italy

**Correspondence:** Alberto Mariani (marianni@alpsolut.eu)

**Abstract.** Recently, Dual-Polarimetric Synthetic Aperture Radar (SAR) has been shown to be effective for large-scale snow cover monitoring, but it faces significant challenges when applied to finer resolutions, which are crucial for applications such as avalanche forecasting. In this study, we propose a novel mathematical model to retrieve snow properties from Sentinel-1 SAR data, leveraging variations in the Dual-Polarimetric Radar Vegetation Index ( $DpRVI_c$ ). We introduce the Snow Index SAR ( $SIsar$ ), which approximates variations in signal depolarization occurring within the snowpack. Our study, conducted in the Central Italian Alps, reveals a strong correlation between the  $SIsar$  index and the snowpack height, enabling accurate snow depth estimation. We also demonstrate the significant impact of the local incidence angle on signal depolarization during the accumulation season. Based on this, we derive a mathematical correction for the incidence angle, whose inclusion in the model reduces snow depth estimation errors by approximately 39 %. The model validation conducted in Tromsø (Norway) and in Davos (Switzerland) confirms its applicability beyond the calibration area, with a root mean squared error (RMSE) of 30.7 cm and a mean absolute error (MAE) of 24.3 cm in Tromsø, and a RMSE of 22.4 cm and a MAE of 18.1 cm in Davos. These findings enhance our understanding of dual-polarimetric Sentinel-1 SAR data sensitivity for high-resolution snow monitoring, providing valuable insights for avalanche forecasting and hydrological applications.

## 1 Introduction

Snow, in addition to being a complex meteorological phenomenon, represents one of the main resources of the mountain environment, and it is well-established that more than one-sixth of the Earth's population relies on glaciers and seasonal snow for their water supply (Barnett et al., 2005). Moreover, this water source is released gradually during the spring and summer season, filling rivers and lakes, allowing the production of hydroelectric power (80 % of Alpine waterways are exploited by hydroelectric power plants) and, above all, renewing the groundwater reserves essential for drinking water supply and agriculture (Soncini and Bocchiola, 2011). Furthermore, snow monitoring is critically important for avalanche forecasting, indeed snow avalanches are among the most significant hazards in mountain regions, causing approximately 100 fatalities in

Europe alone each winter (EAWS, 2025). While for environmental purposes the study of snow at the large-scale is essential (i.e., from a single mountain range to the entire globe), for avalanche forecasting purposes there is more interest in the distribution and properties of the snowpack at the small-scale (i.e., from an individual slope to a single mountain massif; Mariani et al. 25 (2023, 2024)).

In recent years, satellite remote sensing has become a fundamental tool for monitoring snow cover properties at the large-scale, particularly in remote regions lacking monitoring stations (Awasthi and Varade, 2021). One of the most important snow properties that can be monitored using remote sensing techniques is the snowpack height ( $HS$ ). In this study, we propose a new mathematical model for estimating  $HS$  using dual-polarimetric Synthetic Aperture Radar (SAR) data, which corrects biases 30 inherent in previous methods, thereby improving the retrieval of  $HS$  at the small-scale.

Initially, optical satellite data with medium or high resolution, such as Moderate-Resolution Imaging Spectroradiometer (MODIS), Advanced Very High-Resolution Radiometer (AVHRR), and Sentinel-2, served as the primary spaceborne data sources for monitoring snow cover extent and surface properties (Feng et al., 2024). However, optical imaging has significant 35 limitations: it cannot provide information under cloud cover or during nighttime and is sensitive only to the surface of the snowpack. To overcome these limitations, the use of SAR data has been widely explored since the first pioneering studies on microwave interaction with snow reported a certain sensitivity on snow as a volume (Ulaby and Stiles, 1981; Kendra et al., 1998; Tsai et al., 2019).

The interaction of electromagnetic waves with snow is a result of the geometrical structure of the snowpack and of the 40 electromagnetic properties of its single components: air, ice, water vapor, and, when the snow is wet, liquid water. The basic electromagnetic properties are thus the relative dielectric constants of ice and liquid water, and their geometrical distribution in the snow cover (Tiuri et al., 1984; Mätzler, 1987). The presence of a certain liquid water content in the snow strongly influences the amount of backscattered signal, making wet snow retrieval one of the primary applications of SAR related to 45 snow monitoring (Picard et al., 2022). Key contributions in the wet snow retrieval include the works of Nagler and Rott (2000) and Nagler et al. (2016). Additionally, SAR satellites are increasingly used to detect avalanche deposits, providing crucial data for avalanche forecasting (Kapper et al., 2023). The sensitiveness of microwave instrument to snow volume variations also 50 depends on the microwave frequency (Strozzi et al., 1997). At the frequencies commonly used for snow monitoring in SAR missions, such as X-band and C-band, the radar signal penetrates a dry snowpack (i.e., a snowpack without liquid water), reaching the underlying soil and leading to surface backscatter, which predominantly contributes to the total received signal in both co- and cross-polarizations (Wiesmann et al., 2007; Awasthi and Varade, 2021). However, the snow volume tends to 55 depolarize the signal, leading to a certain backscatter increase, compared to bare soil, in cross-polarization (Kendra et al., 1998; Chang et al., 2014). The influence of the underlying ground is more limited at high-frequency (i.e., Ku-band; Tsang et al. (2022)), however, despite this, high-resolution Ku-band satellite products are currently unavailable, and new missions, such as the European Space Agency (ESA) Cold Regions Hydrology High-Resolution Observatory (CoReH2O; Rott et al. (2012)), have not been selected for implementation.

In Pettinato et al. (2013), the potential of COSMO-SkyMed X-band SAR for  $HS$  retrieval was demonstrated, highlighting the capability of a radiative transfer model to simulate snowpack backscatter at this frequency. To overcome the difficulties

in neglecting the underlying soil contribution to the total backscatter at low-frequency SAR platforms, Reppucci et al. (2012) utilized polarimetric decomposition of full-polarimetric RADARSAT-2 products (C-band) to extract the volumetric backscatter contribution and retrieve snowpack characteristics. Finally, it is important to mention works such as the one of Li et al. (2017) 60 or the one of Oveisgharan et al. (2024), in which interferometric SAR (InSAR) technique is used to invert snowpack properties. However, it has been demonstrated that such techniques perform well mainly at longer wavelengths, such as the L-band (Rott et al., 2003). Unfortunately, X-band, L-band, and quad-polarimetric products are currently confined to commercial platforms. Therefore, several studies have focused on exploring the use of publicly available, dual-polarimetric, C-band Sentinel-1 65 products through backscatter-based techniques.

A significant advancement in understanding the response of the alpine snowpack to the Sentinel-1 signal is presented in Brangers et al. (2024), where the use of a tower-based radar system demonstrated that the C-band backscatter generated by the snowpack is not negligible, and can even exceed the backscatter from the underlying ground towards the end of the accumulation season. In Lievens et al. (2019), the separation of ground and snowpack contributions to the total backscatter 70 in dual-polarimetric Sentinel-1 products was achieved by computing the ratio between the cross-polarization channel (VH), which is more sensitive to volumetric backscatter, and the co-polarization channel (VV), which is more sensitive to the surface backscatter produced at the snow-soil interface (formally named depolarization ratio). The resulting index was then used to map  $HS$  at 1 km and 500 m resolution over the European Alps in Lievens et al. (2022). Moreover, Feng et al. (2024) 75 recently demonstrated that the dual-polarimetric radar vegetation index ( $DpRVI_c$ ) outperforms all other dual-polarimetric indices recoverable from Sentinel-1 in  $HS$  retrieval. This index describes an approximation of the degree of depolarization of the backscattered signal, and theoretically ranges from 0, indicating no depolarization, to 1, representing full depolarization of the signal (Mandal et al., 2020). **The  $DpRVI_c$  index is a simplified version of the  $DpRVI$  index, adapted to be computed from a Ground Range Detected (GRD) Sentinel-1 product (Feng et al., 2024).** Full signal depolarization is theoretically obtained when the intensity of the cross-polarized band equals the intensity of the co-polarized band. Since snowpack volumetric backscatter causes signal depolarization, the  $DpRVI_c$  index increases as  $HS$  increases, and its ability in snow depth estimation 80 is demonstrated for the Scandinavian Alps. Further progress in understanding the backscatter mechanisms of the snowpack in response to Sentinel-1's C-band signal, as well as the sensitivity of derived polarimetric indices and interferometric coherence to seasonal snow accumulation, is presented in Jans et al. (2025). The cited study also analyzes the influence of the incidence angle between the SAR signal and the normal to the slope surface, known as the local incidence angle ( $LIA$ ), on  $HS$  and on the snow water equivalent ( $SWE$ ) sensitivity, at a resolution of 1 km, over the Alps.

Despite the significant progress made in leveraging SAR data to monitor snowpack properties at large-scales, limited attention has been given to smaller-scale investigations. Studies targeting specific mountain ranges, valleys, or avalanche forecasting zones where a resolution at the scale of the individual slope is required, are still scarce.

**The aim of this study is to provide a detailed analysis of the capability of the  $DpRVI_c$  index to retrieve snow depth and other snowpack properties at a small spatial scale.** To this end, we compute a snow index, here named Snow Index SAR ( $SIsar$ ), based on  $DpRVI_c$  variations, taking as reference its summer average value. We examine the relationship between the  $SIsar$  index and measured or simulated snowpack properties for two proximal locations in the Central Italian Alps.

The snowpack is a highly dynamic material that undergoes several metamorphic processes throughout the season, altering its physical and mechanical properties (McClung and Schaefer, 2023). For this reason, we try to identify which snowpack variables are more correlated to SAR signal depolarization variations, described by the *SIsar* index. Additionally, the complex mountain topography, combined with the side-looking geometry of SAR, causes significant variation in the *LIA* at the small-scale. We hypothesize that as the *LIA* increases, the SAR signal penetrates more snow, leading to greater depolarization at the slope-scale. Therefore, we analyze in detail the influence of the *LIA* on the *SIsar* index. Based on this analysis, we propose a novel mathematical model to estimate *HS*, which turns out to be the snowpack variable most strongly correlated to our index, demonstrating a significant improvement resulting from considering the *LIA*. The model is validated with in situ observations collected in an area around Tromsø (Norway), which is significantly different from the calibration area in terms of snow and weather conditions, and with a photogrammetric snow depth product near Davos (Switzerland).

## 2 Study areas and data

### 2.1 Study areas

We considered three study areas: a model calibration area and two model validation areas. The model calibration area is located around the municipality of Livigno (Sondrio province, Italy), on the border with Switzerland in the central Rhaetian Alps (46°28' N, 10°8' E). The elevation ranges from 1800 m a.s.l. on the Livigno valley floor to 4049 m a.s.l. at the summit of Piz Bernina. The area experiences an Alpine climate, with snow regimes transitioning from continental in the northern part and maritime in the southern region (McClung and Schaefer, 2023). The Livigno valley floor is predominantly anthropogenic, with a ski area occupying both sides of the valley up to 2800 m a.s.l.. The higher alpine regions are mainly covered by alpine meadows and talus, with widespread permafrost present (Dramis and Gugliemin, 2001). In the southwestern part of the area, numerous alpine glaciers can be found. The intense tourist activity in the area, particularly among winter outdoor enthusiasts, has led to the development of a local avalanche forecasting system, and several automatic weather stations (AWSs) equipped with standard gauges are operational (Monti et al., 2014). The first model validation area is located around Tromsø, Norway (69°38'58" N, 18°57'25" E). This region is characterized by an Arctic transitional snow climate (Velsand, 2017), and features relatively smoother topography compared to the Alpine area used for model calibration. The second model validation area is located near Davos, Switzerland (46°49'20" N, 09°50'02" E) and corresponds to the south-facing slope of Salezerhora peak (2537 m a.s.l.). This region ranges in altitude from 1660 m a.s.l. to 2500 m a.s.l. and is characterized by alpine meadows, with a few small forested sections that have been excluded.

To analyze the evolution of the SAR signals over the seasons and their dependency on the snowpack properties, we selected two sampling areas inside the model calibration area (hereafter referred to as regression sampling areas) around two AWSs: Gessi (46°31'23" N, 10°7'27" E – 2633 m a.s.l.) and Vall (46°28'37" N, 10°11'28" E – 2660 m a.s.l.). Both areas were defined with a 25 m square buffer around the locations of the AWSs, which was adjusted to exclude zones affected by wind deposition and erosion, while aiming to keep the morphology as homogeneous as possible. Both regression sampling areas have slopes ranging from 0° to 12°, with the AWSs located on flat terrain, and a predominant northwest slope orientation. The area

surrounding the Gessi AWS is a typical periglacial environment, characterized by a landscape of scattered stones interspersed with moss and alpine steppe; the presence of permafrost cannot be excluded. In contrast, the area around the Vall AWS exhibits a smaller concentration of stones and is dominated by alpine steppe, interspersed with smooth bedrock outcrops. To analyze the relationship of SAR signal variations with the *LIA*, we selected two larger areas around the location of the AWSs (hereafter referred to as *LIA* sampling areas) to ensure a heterogeneous *LIA* distribution (see Fig. 1). For these areas, a 2 km buffer was  
applied around the location of the AWSs, with elevation differences limited to  $\pm 200$  m. Zones with anthropogenic features, such as ski slopes or zones where snow cover is heavily influenced by ski touring and freeride tracks, were excluded based on experts' experience. The *LIA* sampling areas surrounding the two AWSs display comparable yet heterogeneous land-cover compositions, characterized by alternating zones of alpine steppe, talus deposits, occasional bedrock outcrops, and dispersed large boulders. It is important to note that the regression sampling areas were included within the *LIA* sampling areas, but the  
former represented only a small fraction of the latter. Therefore, despite the partial overlap, the results obtained from analyzing the second area were largely independent from those related to the first, as they were mainly driven by new additional data.

To validate the results, we selected a  $50\text{ km} \times 50\text{ km}$  area inside the Norwegian model validation area where in situ measurements were available. Within this area, we applied a 25 m square buffer around each measurement location to sample the model estimations (hereafter referred to as Norwegian validation sampling areas). Concerning the Swiss validation area, we  
considered an area of approximately 2 km<sup>2</sup> where a photogrammetric snow depth product was available (as will be shown later in the paper, our analysis predominantly focused on snow depth retrieval). Images of the two validation areas will be presented in Sect. 4.3 together with the related results.

## 2.2 SAR data

In the present study, we employed freely available level-1 GRD data products from the Sentinel-1 platform, which provides  
data from a dual-polarization C-band Synthetic Aperture Radar (SAR) instrument at 5.405 GHz. The data were acquired in Interferometric Wide swath mode (IW), with an original resolution of 5 m in range and 20 m in azimuth. Both VV and VH polarization are available and the platform is right-looking. During the SAR data preprocessing phase, as well as to compute the local incidence angle for the two AGs, we used the freely-available Copernicus GLO-30 DEM (digital elevation model) with a 30 m cell resolution (European Union, 2021).

For the calibration of our model we selected two acquisition geometries (AGs) that entirely cover the calibration area, taking advantage of the different acquisition times and look directions. The first AG, with relative orbit number 168, was in the descending direction and was acquired around 5 a.m. UTC. The second orbit, with relative orbit number 15, was in the ascending direction and acquired around 5 p.m. UTC. The different acquisition times provided valuable information for monitoring snowpack conditions, which could vary significantly between morning and afternoon at this latitude. In the present  
study, we focused on the period from October 2022 to July 2024, covering two full snow seasons. Given the Sentinel-1 revisit time of 12 days, a total number of 57 descending plus 59 ascending Ground Range Detected (GRD) products were downloaded in Cloud Optimized Geotiff format (COG) (Copernicus, 2025).

**Figure 1.** General overview of the model calibration area together with the locations of the two AWSs and the *LIA* sampling areas.

For the validation of the model we downloaded Sentinel-1 GRD products over the Norwegian validation area for the months of December 2024 and January 2025. The two AGs had relative orbit numbers 131, with *descending* direction and acquisition time around 5 a.m. UTC, and 95, with *ascending* direction and acquisition time around 5 p.m. UTC. **Concerning the Swiss validation area, we downloaded a Sentinel-1 GRD product acquired on 9 January 2022 at 5 p.m. UTC (AG 15).**

### 2.3 Weather and snowpack measurements

Concerning the calibration area, the weather data utilized to run the snowpack evolution model were collected by the two AWSs, which operated throughout the analysis period. These measurements included air and snow-surface temperature, snowpack height, relative humidity, atmospheric pressure, wind speed and direction, and incoming shortwave radiation. It should be noted that these AWSs are routinely employed for snowpack simulations to support avalanche forecasting within the Livigno municipality (additionally, standard *in situ* snow stratigraphies are routinely performed in the backcountry areas to calibrate model parameters and validate simulation outputs; Monti et al. (2016)). During the 2022-2023 winter season, meteorological conditions were marked by below-average precipitation. The maximum *HS* recorded at the Vall station was 146 cm on 22 April 170 2023. Field observations revealed that the combination of low temperatures and reduced snowpack thickness led to significant constructive metamorphism, resulting in a generally low-density snowpack dominated by depth hoar and faceted grains in the

deeper layers (McClung and Schaefer, 2023). In contrast, the 2023-2024 winter season experienced frequent Atlantic weather systems, leading to above-average temperatures and precipitation. The maximum  $HS$  during this period was 322 cm, recorded at the Vall station on 3 April 2024. These conditions produced a dense and warm snowpack overall.

To the Norwegian validation area are associated 27 in situ snow stratigraphies or snow depth manual observations, already presented by Engeset et al. (2018). These measurements were acquired between the 10 December 2024 and 15 January 2025 in dry snow conditions. On the other hand, to the Swiss validation area is associated a snow depth raster obtained by differentiating a summer digital surface model (DSM) realized with a UAV-photogrammetric survey with another DSM carried out on 12 January 2022 in dry snow condition (Bühler et al., 2022). The dataset has an original resolution of 10 cm. These measurements  
were made three days after the corresponding SAR acquisitions, but the snowpack did not change significantly during that period, so the temporal offset is not expected to strongly affect the comparison.

### 3 Methods

#### 3.1 Snowpack modelling

Snow observations directly measured by the AWSs were integrated with data derived by snowpack simulations. For this purpose, the SNOWPACK software was used (Bartelt and Lehning, 2002), with the operational setting employed for avalanche forecasting purpose in Livigno. The simulations were performed for a flat terrain (no preferential exposition), where the model was forced to follow the snowpack height measured by the AWSs, thus also simulating wind-driven snow erosion. The simulated data had an hourly temporal resolution. For each date and time of the available SAR acquisitions, we extracted the following snowpack variables ( $X_{sv}$ ):

– snow height directly measured by the AWSs ( $HS$ );  
– snow surface temperature directly measured by the AWSs ( $T_{ss}$ );  
– simulated snow water equivalent ( $SWE$ );  
– average simulated snow density of the entire snowpack ( $\rho_s$ );  
– average simulated grain size, referring to the grain diameter, of the entire snowpack ( $E_s$ );  
– average simulated liquid water content of the entire snowpack ( $LWC$ ) expressed in percent by volume;  
– height of the new snow of the last 24 hours ( $HN24$ ).

#### 3.2 SAR data processing

We used the open-source software SNAP for the entire standard preprocessing workflow of the Sentinel-1 GRD products, as well as for extracting the  $LIA$ . Initially, we applied the orbit state vectors provided by the satellite facility and then we

performed radiometric calibration following the beta-nought ( $\beta^0$ ) convention. Next, we removed the radiometric variability associated with topography using the Radiometric Terrain Correction algorithm proposed by Small (2011), obtaining the backscatter coefficient gamma-nought ( $\gamma^0$ ). This coefficient was subsequently subjected to geometric terrain correction and speckle filtering using the Lee filter with a  $9 \times 9$  pixels window size and 2 looks (Yommy et al., 2015). **The original resolution of the GRD products was 10 m.**

For each preprocessed scene we computed the simplified  $DpRVI_c$  index as proposed by Feng et al. (2024):

$$DpRVI_c = \frac{(\gamma_{\text{VH}}^0)^2 + 3\gamma_{\text{VH}}^0\gamma_{\text{VV}}^0}{(\gamma_{\text{VH}}^0 + \gamma_{\text{VV}}^0)^2}, \quad (1)$$

where  $\gamma_{\text{VH}}^0$  represents the backscatter coefficient  $\gamma^0$  in cross-polarization, while  $\gamma_{\text{VV}}^0$  represents the backscatter coefficient  $\gamma^0$  in co-polarization. Both of them are measured in linear scale. Note that for a fixed value of  $\gamma_{\text{VH}}^0$ , the  $DpRVI_c$  index decreases as  $\gamma_{\text{VV}}^0$  increases because  $\partial_{\gamma_{\text{VV}}^0} DpRVI_c = \gamma_{\text{VH}}^0(\gamma_{\text{VH}}^0 - 3\gamma_{\text{VV}}^0)/(\gamma_{\text{VH}}^0 + \gamma_{\text{VV}}^0)^3 

$$X_{sv\_{mdl\_lin}} = \frac{SIsar}{a}, \quad (7)$$

270 where  $X_{sv\_{mdl\_lin}}$  denotes the linear model-based approximation of  $X_{sv}$ . Concerning the study of the dependence of the  $SIsar$  index on the  $LIA$ , we expressed this relationship as:

$$SIsar \simeq \tilde{f}(X_{sv}, LIA) = g(LIA) \cdot X_{sv}, \quad (8)$$

where the function  $g$  was derived again with the least squares method and its shape was chosen according to the distribution of the data derived from the  $LIA$  sampling areas. Indeed, when selecting a time and a  $LIA$  sampling area, we could assume 275 (especially at the beginning of the winter season, when wind redistribution and variations in melting or metamorphism are not very influential) that the snowpack variable  $X_{sv}$  remained constant in that zone. On the other hand, the  $LIA$  took on many different values. For this reason, the function  $g$  could be approximated as  $g(LIA) \simeq SIsar/X_{sv}$ , which should be independent on  $X_{sv}$ . Therefore, the final model for estimating  $X_{sv}$  was:

$$X_{sv\_{mdl\_LIA}} = \frac{SIsar}{g(LIA)}, \quad (9)$$

280 where  $X_{sv\_{mdl\_LIA}}$  denotes the approximation of  $X_{sv}$  depending on the local incidence angle.

### 3.4 Model validation strategy

The validation of our model for approximating  $X_{sv}$  was divided into four parts. Initially, we compared, in terms of root mean squared error (RMSE) and mean absolute error (MAE), the approximations for  $X_{sv}$  given by Eqs. (7) and (9), using as 285 reference the measurements of the two AWSs ( $X_{svmsr}$ ; *msr* stands for measured) inside the model calibration area. Secondly, we performed a mathematical analysis to establish the validity of the  $LIA$  dependency through the function  $g$ . When we initially derived the value  $a$  we considered values of the  $SIsar$  index related to different  $LIA$ s. For this reason, the weighted average value  $\bar{g}$  of the function  $g$  should be similar to  $a$ . Note that we had to consider a weighted average because some  $LIA$ s were more recurring than others. Recall that the regression sampling areas and the  $LIA$  sampling areas were effectively 290 independent of each other. Thirdly, using Eq. (9), we computed the values of  $X_{sv\_{mdl\_LIA}}$  for the Norwegian validation sampling areas. To demonstrate the model's applicability in a location significantly different from the one it was calibrated on, we compared  $X_{sv\_{mdl\_LIA}}$  with field measurements conducted in the Norwegian validation sampling areas. Finally, using again Eq. (9), we computed the values of  $X_{sv\_{mdl\_LIA}}$  for the Swiss validation area. Those values were compared to the photogrammetric data to validate the model with a large dataset.

## 4 Results

### 295 4.1 *SIsar* index and snowpack variables

Due to the limited dataset available, the results of the random forest analysis are exploratory. The importance values of the variables are uniformly low and, therefore, not reported, as they are likely affected by the small sample size and should not be overinterpreted. Nevertheless, the random forest model suggests that the variable  $HN24$  is not important for predicting the *SIsar* index, while  $HS$  emerged as the most important predictor, followed by  $\rho_s$ ,  $E_s$ ,  $LWC$ , and  $T_{ss}$ . The results of the 300 statistical analysis, conducted on the entire dataset (which includes all regression sampling areas and acquisition geometries), are presented in Table 1. The analysis reveals significant correlations between the *SIsar* index and several variables, including  $HS$ ,  $SWE$ ,  $\rho_s$ , and  $E_s$ . Specifically, there is a moderate-to-strong positive relationship with both  $HS$  and  $SWE$ , a moderate positive correlation with  $\rho_s$ , and a moderate-to-low negative correlation with  $E_s$ . These findings align with the results from the 305 random forest model. The PCCs and the related  $p$ -value tests indicate that the linear relationships between the *SIsar* index and both  $HS$  and  $SWE$  are strong, while no robust linear relationship is found between the *SIsar* index and  $\rho_s$ . Moreover, the  $p$ -values associated to the linear regression model show that, for both  $HS$  and  $SWE$ , the regression slope is significant. However, only for  $HS$  the intercept is not significant. Therefore, only for  $HS$ , we can set the intercept to zero, implying that 310 snow-free conditions should correspond to a value of  $SIsar \simeq 0$  (see Sect. 3.3). This, together with the fact that the best PCC corresponds to  $HS$ , supports the selection of this variable as the candidate  $X_{sv}$  for the model. Therefore, hereafter we set  $X_{sv} = HS$ . In particular, the quality of the linear regression with intercept set equal to zero is almost the same as the one obtained for a generic intercept, indeed the  $R^2$  coefficients are 0.493 and 0.495 respectively. The coefficient  $a$  of the linear relationship between  $HS$  and the *SIsar* index (see Eq. (6)) is approximately  $6.00 \cdot 10^{-4} \text{ cm}^{-1}$ , as reported in Fig. 5(a).

The results of the statistical analysis conducted for each individual regression sampling area are not presented in tabular form for brevity. However, the findings are consistent with those reported for the analysis conducted on the entire dataset. 315 Focusing only on the most influential variables, a strong positive PCC with the *SIsar* index is observed for  $HS$  and  $SWE$  across all areas and AGs. In particular, the highest PCC values for  $HS$  and  $SWE$  are at Vall for AG 15, with PCC equal to 0.802 and 0.800, respectively. Additionally, the correlation intensity order among the variables is in line with the results from the analysis of the entire dataset, except for the Gessi area in AG 168, where  $SWE$  shows a slightly better linear correlation with the *SIsar* index than  $HS$  (PCC equal to 0.761 against 0.756). Finally, the best correlations are in general related to the 320 AG 15 (afternoon).

### 4.2 *SIsar* index and local incidence angle

Figure 2 shows examples of the behavior of the *SIsar* index for  $5^\circ$  *LIA* classes across the different *LIA* sampling areas, at different times during the snow season, and under snow-free conditions. In both areas, for all acquisitions with snow-free conditions, the *SIsar* index on average remains approximately constant as the *LIA* increases, with values close to zero. When 325 analyzing the relationship in both *LIA* sampling areas for all dates with snow cover, we first observe that the *SIsar* index shows negative values for angles approximately smaller than  $30^\circ$ , even if  $HS$  is not zero. For all these acquisitions, for angles

**Table 1.** Summary of the results of the statistical analysis for each snowpack variable  $X sv$ . The linear regression is performed only for those variables showing a significant and large PCC.

| Snowpack variable<br>( $X sv$ ) | SCC    | SCC<br><i>p</i> -value | PCC    | PCC<br><i>p</i> -value | Linear regression<br>intercept <i>p</i> -value | Linear regression<br>slope <i>p</i> -value |
|---------------------------------|--------|------------------------|--------|------------------------|------------------------------------------------|--------------------------------------------|
| $HS$                            | 0.552  | $2.60 \cdot 10^{-9}$   | 0.704  | $3.23 \cdot 10^{-16}$  | 0.485                                          | $3.23 \cdot 10^{-16}$                      |
| $SWE$                           | 0.553  | $2.43 \cdot 10^{-9}$   | 0.700  | $5.11 \cdot 10^{-16}$  | 0.00761                                        | $5.11 \cdot 10^{-16}$                      |
| $\rho_s$                        | 0.438  | $6.43 \cdot 10^{-9}$   | 0.492  | $2.07 \cdot 10^{-7}$   | -                                              | -                                          |
| $E_s$                           | -0.295 | 0.00302                | -0.275 | 0.00566                | -                                              | -                                          |
| $T_{ss}$                        | 0.0367 | 0.717                  | 0.0682 | 0.500                  | -                                              | -                                          |
| $HN24$                          | 0.199  | 0.0471                 | 0.155  | 0.124                  | -                                              | -                                          |
| $LWC$                           | 0.349  | $3.72 \cdot 10^{-4}$   | 0.139  | 0.168                  | -                                              | -                                          |

greater than approximately  $30^\circ$ , the relationship between the  $SIsar$  index and the  $LIA$  follows a parabolic shape, which suggests a relationship between them in the presence of snow.

As one can notice from Fig. 3, the point where the curves showing the relation between the  $LIA$  and the  $SIsar$  index intersect the  $x$ -axis is consistently around  $25^\circ - 30^\circ$ , and again we see that the relationship between the  $SIsar$  index and the  $LIA$  follows a parabolic shape for larger angles. In particular, as  $HS$  increases the parabolic shape expands vertically upwards. This suggests that the  $SIsar$  index can be expressed as Eq. (8). Figure 3 also reports the behaviors of the  $DpRVI_c$  index and of the single backscatter components  $\gamma_{VH}^0$  and  $\gamma_{VV}^0$  as the  $LIA$  varies. Under snow conditions, the  $DpRVI_c$  index attains values smaller than under snow-free conditions only for  $LIA$ s smaller than  $25^\circ - 30^\circ$ , in line with the  $SIsar$  index behavior. Unlike the latter, its values related to the summer acquisition depend on the  $LIA$ . Note that  $\gamma_{VV}^0$  increases with  $HS$  for  $LIA$ s below approximately  $20^\circ$ , in line with the findings of Jans et al. (2025), while it appears to be independent on  $HS$  for larger  $LIA$ s. However, it decreases as the  $LIA$  varies between  $20^\circ$  and  $80^\circ$ . On the other hand,  $\gamma_{VH}^0$  shows a more complex behavior as the  $LIA$  varies. Interestingly, for  $LIA$  values below  $30^\circ$ , it reaches smaller values under snow-covered conditions than under snow-free conditions. However, we observe the opposite trend for larger  $LIA$ s and a positive correlation with  $HS$  in line with the results of Brangers et al. (2024).

From the least squares method, we obtain the following equation for  $g(LIA)$  describing a parabola, as represented in Fig. 4:

$$g(LIA) = a_0 + a_1 \cdot LIA + a_2 \cdot LIA^2, \quad (10)$$

where the  $LIA$  is expressed in degrees,  $a_0 = 4.41 \cdot 10^{-3} \text{ cm}^{-1}$ ,  $a_1 = 2.04 \cdot 10^{-4} \text{ }^\circ\text{cm}^{-1}$ , and  $a_2 = -1.80 \cdot 10^{-6} \text{ }^\circ\text{cm}^{-1}$ . The resulting  $R^2$  coefficient is equal to 0.695. The zeros of  $g(LIA)$  are approximately  $29.1^\circ$  and  $84.4^\circ$ . Since we observe that for  $LIA$  values below approximately  $30^\circ$  the  $SIsar$  index is always negative, even when  $HS$  is nonzero, we exclude all areas with  $LIA 

**Figure 2.** Box plots of  $5^\circ$  *LIA* classes vs. *SIsar* index values over the course of the snow accumulation season (i.e., for increasing values of *HS* measured by the AWSs within the *LIA* sampling areas). (a) Gessi AG 15, 8 August 2023, snow-free conditions. (b) Gessi AG 168, 23 November 2023,  $HS_{msr} = 60$  cm. (c) Gessi AG 15, 24 November 2023,  $HS_{msr} = 60$  cm. (d) Vall AG 168, 29 December 2023,  $HS_{msr} = 100$  cm.

are related to the *LIA*; see Sect. 3.2). Areas with  $LIA > 80^\circ$  are also removed because they are in shadow. Note that, due to these choices, Eq. (9) is well-defined since  $g(LIA) \neq 0$  for all *LIA*s of interest.

**Figure 3.** Mean  $SI_{SAR}$  (a),  $DpRVI_c$  (b),  $\gamma_{VH}^0$  (c), and  $\gamma_{VV}^0$  (d) values, sampled from the  $LIA$  sampling areas, grouped into  $LIA$  classes of  $2^\circ$  width. The blue color scale represents the increasing  $HS$  values measured by the two AWSs for different acquisition dates along the 2023-2024 season. Yellow points represent the values from an acquisition date in summer 2023. The legend in (b) is the same for all subfigures.

**Figure 4.** Mean  $SI_{SAR}$  values normalized by  $HS$  measured by the AWSs grouped into  $LIA$  classes of  $2^\circ$  width. Data derive from the beginning of the season, when we could assume that  $HS$  was representative of the entire  $LIA$  sampling areas. Values for  $LIA < 30^\circ$  and  $LIA > 80^\circ$  are reported in yellow, while values between  $30^\circ$  and  $80^\circ$  in red. In black we report the function  $g(LIA)$ , inferred on the red data.

**Figure 5.** (a) Scatter plot of  $HS_{msr}$  vs.  $SI_{sar}$ . (b) Scatter plot of  $HS_{msr}$  vs.  $HS_{mdl\_LIA}$ . The linear regressions computed with the least square method are also represented.

**Table 2.** Errors and number of data points related to the different models and datasets.

| HS estimation                        | RMSE    | MAE     | number of data points |
|--------------------------------------|---------|---------|-----------------------|
| $HS_{mdl\_lin}$ (Livigno, Italy)     | 57.2 cm | 46.0 cm | 100                   |
| $HS_{mdl\_LIA}$ (Livigno, Italy)     | 34.6 cm | 28.3 cm | 100                   |
| $HS_{mdl\_LIA}$ (Tromsø, Norway)     | 30.7 cm | 24.3 cm | 27                    |
| $HS_{mdl\_LIA}$ (Davos, Switzerland) | 22.4 cm | 18.1 cm | 589                   |

### 350 4.3 Models comparison and validation

When applied to the calibration area, the model for  $HS$  given by Eq. (7) leads to a RMSE equal to 57.2 cm and a MAE equal to 46.0 cm. With the *LIA* correction, the errors drop significantly, indeed using Eq. (9) the RMSE becomes 34.6 cm while MAE 28.3 cm. These values represent a 39.5 % decrease in RMSE and a 38.5 % decrease in MAE. Plotting  $HS_{mdl\_LIA}$  against  $HS_{msr}$  (Fig. 5(b)), we observe that  $HS_{mdl\_LIA} \simeq c_1 \cdot HS_{msr}$ , with  $c_1 = 0.841$ . It is important to note that a perfect 355 model would lead to a value of  $c_1$  equal to 1. To mathematically verify the validity of  $g(LIA)$  we compute its weighted average  $\bar{g}$  as explained in Sect. 3.4, obtaining  $\bar{g} = 9.70 \cdot 10^{-4} \text{ cm}^{-1}$ , which is of the same order of magnitude as the coefficient  $a = 6.00 \cdot 10^{-4} \text{ cm}^{-1}$  of Eq. (7). Moreover, using Eq. (9) to estimate  $HS$  in the validation area around Tromsø (the Norwegian validation dataset; see Fig. 6(b)) we obtain an RMSE of 30.7 cm and a MAE of 24.3 cm. Finally, using Eq. (9) to estimate  $HS$  in the validation area around Davos (the Swiss validation dataset; see Fig. 7) we obtain an RMSE of 22.4 cm and a MAE of 360 18.1 cm. See Table 2 for a comparison of the errors and the number of data points.

**Figure 6.** Snow depth estimation for the Norwegian validation area. (a) Overview of  $HS_{mdl,LIA}$  between 31 December 2024 and 1 January 2025. (b)  $HS_{msr}$  vs.  $HS_{mdl,LIA}$  related to the 27 in situ snowpack observations. (c) Particular of (a) showing in situ observations carried out between 30 December 2024 and 2 January 2025 (no significant variation of  $HS$  occurred in this time window).

## 5 Discussion

### 5.1 Correlations of the *SIsar* index with snowpack variables

The statistical analysis indicates that the *SIsar* index is a complex function of several snowpack parameters. Nevertheless, since SAR measurements are of great interest, especially in poorly monitored areas where no additional observations are 365 generally available, we aimed to limit the dependency to just one of the most significantly linearly correlated variables. These are  $HS$  and  $SWE$ , in accordance with several other studies (Snehmani et al., 2015; Lievens et al., 2019; Patil et al., 2020). The selection of  $HS$  as candidate  $X_{sv}$  is supported by the fact that it is the only snowpack variable for which the intercept value is not statistically significant, thereby fulfilling the theoretical condition that  $SIsar \simeq 0$  corresponds (on average) to a snow-free state.

370 The Spearman's correlation coefficient suggests that the *SIsar* index increases also with  $\rho_s$ . One possible explanation for this increase is that thicker and denser snowpacks contain larger snow mass and more grains per unit volume of snow, consequently, there will be more opportunities for signal depolarization, as reported in Lievens et al. (2019). In addition, Tiuri et al. (1984) demonstrated that both the real and imaginary parts of the dry-snow dielectric constant increase with snow density. As this happens, the surface backscatter enhances, while the influence of the underlying soil may be slightly reduced due to

**Figure 7.** Swiss validation area. (a) Difference between  $HS_{mdl\_LIA}$  and  $HS_{photo}$ , derived from the photogrammetric survey. (b)  $HS_{mdl\_LIA}$  vs.  $HS_{photo}$ . (c)  $HS_{photo}$  (left) and  $HS_{mdl\_LIA}$  (right).

375 the stronger absorption. However, the Pearson's correlation coefficient suggests that the relation between the  $SIsar$  index and  $\rho_s$  is not linear. This is in line with the linear relationship resulted between the  $SIsar$  index and the  $SWE$ : if both  $\rho_s$  and  $HS$  would be linearly correlated with the index, then the  $SWE$  would be quadratically correlated with it (this directly follows from the definition of  $SWE$ ; see Sect. 3.3). Concerning the fact that the intercept can be set equal to zero for  $HS$  but not for the  $SWE$ , this is probably a consequence of the fact that the  $SWE$  is not well-defined in absence of snow: as snow fades away, both its mass and volume tend to zero, implying that  $\rho_s$  cannot be defined in absence of snow. A backscatter increase with  $\rho_s$  is also observed in Besic et al. (2012), who however demonstrate how the  $\rho_s$  influence on SAR-based  $SWE$  retrieval is negligible.

380 Paloscia et al. (2017) observed that the co-polarized band of Cosmo-SkyMed  $\sigma_{HH}^0$  increases with larger  $E_s$ . Assuming that  $\gamma_{VV}^0$  behaves similarly to  $\sigma_{HH}^0$ , this could explain the slight decrease in the  $SIsar$  index that we observe for increasing  $E_s$ . Nevertheless, further analysis are required to understand the influence of this parameter. Interestingly, the results do not show a significant correlation between the  $SIsar$  index and  $T_{ss}$  even if Baumgartner et al. (1999) demonstrated that SAR backscatter is sensitive to variations in near-surface snow temperature.

390 Finally, it should be emphasized that the  $LWC$  exhibits one of the weakest correlations with the  $SIsar$  index, despite its primary role in the snow backscattering mechanism (Tiuri et al., 1984; Nagler and Rott, 2000; Besic et al., 2012). However, this is not surprising since we excluded all acquisitions containing wet snow according to the Nagler's method, which obviously significantly reduced the variability of the  $LWC$ .

The statistical analysis conducted for individual regression sampling areas and individual acquisition geometry confirm the one conducted on the entire dataset. The evidence that the Pearson's correlation coefficients for individual stations and acquisition geometry slightly differ from the one obtained when considering all data together suggests that variations in morphology, 395 land cover, and local climatic conditions lead to slight differences in the relationship between the *SIsar* index and any  $X_{sv}$ . Consequently, a local fitting approach should perform slightly better than a global one. These results are consistent with the findings of Jans et al. (2025).

## 5.2 Influence of the local incidence angle on the *SIsar* index

The analysis of the relationship between the  $DpRVI_c$  index and the *SIsar* index with the *LIA* under snow-free and snow- 400 covered conditions reported in Fig. 3 is revealing. In particular, the findings regarding the two single backscatter components  $\gamma_{VH}^0$  and  $\gamma_{VV}^0$  are in line with the results presented by Brangers et al. (2024) and Jans et al. (2025), respectively. The increase of  $\gamma_{VH}^0$  as *HS* increases can be explained by the larger volume of snow traversed by the SAR signal, which leads to more opportunities for signal depolarization and by the different interactions with layered structures. The fact that  $\gamma_{VV}^0$  shows no 405 dependence on *HS* indicates that it is primarily influenced by soil properties, further confirming the effectiveness of using depolarization ratios to isolates the snow backscatter contributions. Note, however, that the variations of  $\gamma_{VH}^0$  with *HS* is very 410 small, attaining values very close to those observed under snow-free conditions. This may explain why Strozzi et al. (1997) concluded that *HS* could not be monitored with C-band SAR due to the low sensitivity of the backscatter components to snow accumulation.

Interestingly, we observe that in presence of snow the  $DpRVI_c$  index attains values significantly larger than those under 415 snow-free conditions for *LIA*s above  $40^\circ$ , and its variations with *HS* is substantial. This confirms that the use of polarization ratios or similar indices significantly increases the sensitivity of SAR to *HS* variations, enabling its monitoring (Lievens et al., 2019; Feng et al., 2024; Jans et al., 2025).

Focusing on the *SIsar* index, no significant dependence on the *LIA* is observed for any of the snow-free acquisitions in either analysis area, with values on average around zero. On the other hand, in all acquisitions under snow-cover conditions, 420 we first observe that *SIsar* values are negative for low incidence angles, specifically for *LIA* values below approximately  $25^\circ - 30^\circ$ . For incidence angles greater than  $30^\circ$ , the relationship between our index and the *LIA* takes on a parabolic shape, increasing up to around  $55^\circ$  before decreasing again. This suggest that  $DpRVI_c$  index variations in presence of snow above the ground are significantly influenced by the *LIA*, and that the latter must be taken into account for dual-polarimetric SAR retrieval of *HS* (despite the SAR products were radiometrically terrain corrected for the *LIA* within the preprocessing phase). A complex relationship between SAR derived vegetation indexes and *LIA* has been already demonstrated in the presence of vegetation cover above the ground by Kaplan et al. (2021). Since the influence of a vegetation volume on the  $DpRVI_c$  can be linked to the one snow over bare soil (Feng et al., 2024), we believe our observations are valid. As shown in Fig. 2, the presence of outliers in the *SIsar* index values across the *LIA* classes is notable. However, such variability is common in this type of experiment and can be attributed to various sources of error, including wind redistribution effects, speckle noise, and 425 spatial variability in snowpack or ground properties.

Since the  $SIsar$  index is consistently negative during the accumulation season for  $LIA$  values smaller than  $30^\circ$ , snow cover monitoring is not possible in this situation. The negative values of the  $SIsar$  index at low  $LIA$ s can be primarily attributed to the increase of  $\gamma_{VV}^0$  with  $HS$  (recall Fig. 3(d)), consistent with previous observations (Jans et al., 2025), which results in snow-covered  $DpRVI_c$  index values lower than those under snow-free conditions. The decrease of the  $SIsar$  index values for 430 large  $LIA$ s is more difficult to interpret. Some possible explanations include a combination of factors such as the dominant specular scattering mechanism, many pixels being in shadows, different interactions with soil and snowpack structures, and travel path of the signal through the snow being so long that its energy absorption is no longer negligible, even at C-band (Rott et al., 2021). However, further specific studies are needed to confirm or refute these hypotheses.

All these considerations, together with the fact that the parabolic curve describing the dependence of the  $SIsar$  index on the 435  $LIA$  extends upwards as the accumulation season progresses (i.e., as  $HS$  increases), support the choice of Eq. (8), and hence also the estimation for  $HS$  given by Eq. (9). Note that analogous relationships between the  $DpRVI_c$  index and  $HS$  cannot be established, due to the index's sensitivity to the  $LIA$  under snow-free conditions (see Fig. 3(b)). Finally, the similar behaviors of the  $SIsar$  index normalized by the measured  $HS$  of both  $LIA$  sampling areas at different times of the snow season (see Fig. 4) suggest that  $g(LIA)$  is not significantly dependent on local or temporal conditions. Notice that acquisitions refer to two 440 very different seasons in terms of structure and physical properties of the snowpack.

### 5.3 Effectiveness of considering the local incidence angle within the model

When we approximate  $HS$  through Eq. (9) (i.e.,  $HS_{mdl\_LIA}$ ) we obtain a significant improvement with respect to the usage 445 of Eq. (7) (i.e.,  $HS_{mdl\_lin}$ ), both in terms of root mean square error and mean absolute error. Furthermore, the values of the weighted average  $\bar{g}$  of the function  $g(LIA)$  is of the same order of magnitude of the slope coefficient  $a$  of the linear regression. Differences between the values of  $\bar{g}$  and  $a$  are attributed to the several sources of errors: we consider only the most influential 450 snowpack variable, variations in soil properties that we do not consider could affect the values of the  $SIsar$  index, and the datasets used to derive  $a$  and  $g(LIA)$  are different and effectively independent of each other (in particular, the former is much smaller than the latter, which naturally results in a smaller variation of  $LIA$ s). This also explains why  $HS_{mdl\_LIA} \simeq c_1 \cdot HS_{msr}$  with  $c_1$  not exactly equal to 1: the slope  $c_1$  incorporates the errors deriving from the factors just described. However, the results 455 obtained not only confirm the significance of the relationship between the  $SIsar$  index and the  $LIA$ , but also indicate that correcting the index with a function accounting for its dependence on the  $LIA$  leads to a substantial improvement. Therefore, the  $LIA$  has a non-negligible influence on SAR signal depolarization when snow is present above the ground.

The results of the validation performed with the in situ measurements carried out in Tromsø, Norway, and Davos, Switzerland, further support the correctness of the modelling. The relation  $HS_{mdl\_LIA} \simeq HS_{msr}$  is successfully verified, and both 455 the RMSE and the MAE are slightly smaller than the corresponding values obtained for Livigno, Italy. This suggests a global applicability of the model, even if a local calibration with in situ observations is recommended, as already discussed in Sect. 5.1. As shown in Fig. 6, mapping  $HS_{mdl\_LIA}$  for the Norwegian validation area reveals a high degree of heterogeneity in snow cover distribution. At a large-scale, we observe that zones with greater snow accumulation are generally found at higher elevations (see Fig. 6(a)). Conversely, at a small-scale, we observe that the snow distribution appears highly irregular, with eroded

460 peaks and ridges, while significant deposits are found in bowls and gullies (see Fig. 6(c)). This latter observation aligns with both field measurements available for the period under analysis and the regional climate, which is characterized by frequent wind transport events. It is important to note that, within the same acquisition, we used field observations that were proximal to one another (minimum spacing of approximately 100 m) but, being the slopes naturally non-homogeneous, they correspond to different *LIAs*. This implies that, although the obtained MAE was slightly larger than the one obtained in the work of 465 Lievens et al. (2022), where  $HS$  was mapped for a 500 m resolution, or in the one of Feng et al. (2024), the proposed model demonstrates a good ability in describing snow depth variations at the small-scale (i.e., within the same slope). This is particularly relevant for avalanche forecasting purposes, since avalanche forecasters are interested in detecting the complex snow cover heterogeneity of the alpine environment, which is influenced by wind transport, avalanches, and the various metamorphic processes occurring at different aspects (Plattner et al., 2004).

470 The quality of snow depth retrieval at the slope scale is confirmed by the validation in Switzerland, where the  $HS$  mapped with our model was compared with measurements derived from a photogrammetric survey. Indeed, the RMSE and the MAE were even smaller than the ones related to the Norwegian validation area. From Fig. 7(a), it is evident that the largest overestimations are concentrated in the upper part of the mountain slope, where the steepest gradients occur and the snowpack may have experienced variations (e.g., snow creep) during the three days between the Sentinel-1 GRD acquisitions and the 475 UAV-photogrammetric survey. On the other hand, the largest underestimations are found in the lower part of the slope, where the snowpack is very thin.

#### 5.4 Model limitations and future work

A key limitation of our model is that the *SIsar* index depends not only on the snow height  $HS$  but also on other snowpack properties, such as the average snow density  $\rho_s$  and average grain size  $E_s$ . In mountain regions, these properties can vary 480 considerably since they are strongly influenced by both elevation and aspect. This variability may result in areas where the model performs slightly better and others where its performance is somewhat reduced.

As reported in Appendix A, we observe that a significant overestimation of  $HS$  when wet snow layers are embedded within dry snow layers. This configuration is typical during the early or late stages of the snow season, particularly when dry snowfall accumulates over an existing wet snow surface due to a decrease in frost level during a precipitation event (Colbeck, 1982). 485 This overestimation can be theoretically interpreted as follows: both  $\gamma_{VV}^0$  and  $\gamma_{VH}^0$  should drop in presence of wet snow since the signal is absorbed and hence unable to penetrate further in depth. However, as observed from analyzing the data, the presence of dry snow on the surface results in a  $\gamma_{VH}^0$  increase, which is more sensitive to volumetric backscatter than  $\gamma_{VV}^0$ . This implies a significant increase of the  $DpRVI_c$  index since it increases as the difference between co- and cross- polarization backscatter decreases. Under these conditions, the method of Nagler et al. (2016) used for SAR-based wet snow retrieval may fail since it 490 relies on both VV and VH bands. Notably, in these conditions, the average liquid water content ( $LWC$ ) of the snowpack is not particularly large, considering that an  $LWC$  threshold of 1 % is commonly used to distinguish dry from wet snow (Mitterer et al., 2013).

The variations of  $HS_{mdl\_LIA}$  in bare-soil conditions reflect the dependence of  $DpRVI_c$  by external factors, such as soil moisture, ground surface temperature, and variations in low-lying vegetation (Das and Pandey, 2024). This confirms that the 495 method is inherently unable to clearly discriminate between the presence and absence of snow, as noted in Jans et al. (2025). Indeed,  $SIsar \simeq 0$  in snow-free conditions (see Sect. 3.3) is verified only on average, not for the single acquisition (see Appendix A). Moreover, even if further studies are needed to confirm this behavior, the **large** number of pixels with negative  $SIsar$  values at the beginning of the snow season confirms what reported in Lemos and Riihelä (2024), who demonstrates the difficulties of Sentinel-1 based  $HS$  retrieval in very thin snowpacks.

Since the model was calibrated with snowpacks not deeper than 300 cm, its validity for thicker snow accumulations remains untested. Future studies will address these gaps, particularly focusing on  $DpRVI_c$  variations in snow-free conditions and explore potential model adjustments for deeper snowpacks, also considering that, theoretically, the signal absorption by Sentinel-1 may become non-negligible for very thick snow layers. Moreover, our study focused on alpine terrain, suggesting that the quality of snowpack height estimations under different terrains and soil covers should be assessed.

When mapping  $HS$  over the validation area near Davos (Switzerland), we observed the presence of few patchy outliers. These anomalous pixels likely resulted from the sensitivity of the  $SIsar$  index to local variations in snow cover conditions or changes in soil properties between the snow-covered and summer reference acquisitions. However, these outliers could be effectively reduced by applying a median filter over a small pixel window (see Sect. 3.2). We therefore recommend using this filter as a post-processing step.

Finally, it is known that the presence of melt-freeze crusts within the snowpack can strongly affect both co- and cross-polarized backscatter coefficients (Brangers et al., 2024), and thus also the performance of our model.

## 6 Conclusions

In this study, we presented a novel mathematical model that enhances small-scale snow depth monitoring leveraging dual-polarimetric Sentinel-1 SAR data. Specifically, we introduced a new index, named Snow Index SAR ( $SIsar$ ), which is defined 515 as the difference between the Dual Polarimetric Radar Vegetation Index ( $DpRVI_c$ ) computed under snow-covered conditions and average snow-free conditions. The model was calibrated using two independent datasets from the Livigno area (Italy, Central Italian Alps), which include data collected from two proximal automatic weather stations and simulated with the SNOWPACK snow cover model. These datasets span two winter seasons and cover two different acquisition geometries.

A statistical analysis revealed that the  $SIsar$  index is influenced by several snowpack variables. In particular, it shows 520 a linear correlation with the snowpack height and the snow water equivalent, while statistically significant relationships were also found with the average snowpack density and grain size. The  $SIsar$  index generally increases with these quantities, except for the average grain size, where a low negative correlation was observed. The strongest linear correlation is with the snowpack height, so we initially applied a linear regression model to estimate snow depth from  $SIsar$  values. Notably, the model aligns well with theoretical expectations, confirming that  $SIsar \simeq 0$  corresponds (on average) to snow-free conditions.

A key result of this study is the demonstration of the strong influence of the local incidence angle (*LIA*) on the *SIsar* index in presence of snow, despite the SAR products are radiometrically terrain corrected for the *LIA* within the preprocessing phase. For angles below 30°, the *SIsar* index is almost always negative, a behavior we explained theoretically. Between 30° and 80°, its relationship with the *LIA* follows a parabolic curve. The derivation of this curve allowed us to correct for the angular dependence of signal depolarization for increasing snowpack height. This correction significantly improved snowpack  
height estimations compared to the initial linear model, reducing the mean absolute error (MAE) by 38.5 % and the root mean square error (RMSE) by 39.5 %.

The final model was verified and validated using two independent dataset from field observations acquired in Tromsø (Norway) and a photogrammetric snow depth products realized near Davos (Switzerland). The validation results showed a RMSE of 30.7 cm and a MAE of 24.3 cm in Tromsø, and a RMSE of 22.4 cm and a MAE of 18.1 cm in Davos, providing preliminary  
evidence of the model's potential for global applicability. In addition to the dependence of our index on several snowpack properties, an additional analysis identified the summer  $DpRVI_c$  index variations and the presence of wet snow layers inside the snowpack as significant sources of error.

## Appendix A: Wet snow and summer *SIsar* index-variations

To further investigate the results, we compared the modelled snowpack height (i.e.,  $HS_{mdl\_LIA}$ ) with the SNOWPACK simulated stratigraphies on the two regression sampling areas and across two seasons (2022/23 and 2023/24). We observed that the average liquid water content (*LWC*) was nonzero, albeit very low, on certain days when Nagler's method did not indicate the presence of wet snow. Looking at the stratigraphies, we noticed that in these dates there were wet snow layers embedded within dry snow layers. As illustrated in Fig. A1 for Gessi with AG 15, overall, the  $HS_{mdl\_LIA}$  exhibits larger fluctuations compared to  $HS_{msr}$ . Moreover, our model seems to significantly overestimate the snowpack height in the situation mentioned  
above.

Furthermore, we assessed the model's capability to distinguish between snow-free and snow-covered conditions. In Fig. A1 are shown the values of  $HS_{mdl\_LIA}$  during the summer 2023 computed for the Gessi regression sampling area. During the observed summer period, the RMSE was 30.5 cm and the MAE 25.6 cm, consistent with snow-covered seasons. Negative *SIsar* values were common and expected, as even small model errors can yield negative estimates when snow depth is zero;  
this behavior is not related to the *LIA* (see Sects. 3.2 and 4.2). Due to the modelled *HS* fluctuations, it is not possible to detect snow-free conditions with our model. Anyway, on average  $HS_{mdl\_LIA} \simeq 0$ , which is in line with the results presented in Sect. 4.2. Similarly, we noticed the presence of pixels characterized by negative *SIsar* values under snow-covered conditions when the snowpack is very thin.

*Data availability.* The data measured by the Vall AWS are available at <https://www.arpalombardia.it/temi-ambientali/meteo-e-clima/form-richiesta-dati/> (ARPA Lombardia, 2025). The data measured by the Gessi AWS and the snow stratigraphies of the Italian model calibration area are available  

**Figure A1.** Comparison between  $HS_{mdl\_LIA}$  (solid red line) and  $HS_{msr}$  (solid black line; Gessi AWS) during the two full seasons 2022/23 and 2023/24. The dotted black lines are obtained by shifting the solid one upward and downward by 31.0 cm, which is the RMSE obtained by estimating  $HS$  considering only the Gessi data. The bar chart represents the liquid water content ( $LWC$ ).

from the corresponding author. The in situ snow stratigraphies of the Norwegian model validation area are freely available at <https://regobs.no/?SelectedNumberOfDays=3&&NWLat=72.47527631092942&NWLon=-21.621093750000004&SELat=55.178867663282006&SELon=89.384765625> (Varsom Regobs, 2018). The photogrammetric data related to the Swiss validation area are available at <https://www.doi.org/10.16904/envidat.376> (Bühler et al., 2022). The SAR data have been downloaded from Copernicus (2025).

*Author contributions.* AM: Conceptualization, Investigation, Formal analysis, Methodology, Validation, Visualization, Writing (original draft preparation). JB: Formal analysis, Methodology, Validation, Visualization, Writing (original draft preparation). FL: Supervision, Writing (review and editing). GV: Data curation. MM: Data curation. FM: Supervision, Writing (review and editing).

*Competing interests.* The authors declare that they have no conflict of interest.

*Acknowledgements.* The authors are grateful to Martin Ahrland Stefan for the discussion regarding the Norwegian validation dataset, and to Yves Bühler for providing the access to the Davos validation dataset. AM acknowledges Alpolut S.r.l. for fully funding his PhD. JB is a member and acknowledges the support of *Gruppo Nazionale di Fisica Matematica* (GNFM) of *Istituto Nazionale di Alta Matematica* (INdAM). JB also thanks the support of the project PRIN 2022 PNRR "Mathematical Modelling for a Sustainable Circular Economy in Ecosystems" (project code P2022PSMT7, CUP D53D23018960001) funded by the European Union - NextGenerationEU, PNRR-M4C2-I 1.1, and by MUR-Italian Ministry of Universities and Research.

ARPA Lombardia, Vall dataset: <https://www.arpalombardia.it/temi-ambientali/meteo-e-clima/form-richiesta-dati/>, last access: 13 January 2025.

Awasthi, S. and Varade, D.: Recent advances in the remote sensing of alpine snow: a review, *GIsci. Remote Sens.*, 58, 852—888, <https://doi.org/10.1080/15481603.2021.1946938>, 2021.

Barnett, T.P., Adam, J.C., and Lettenmaier, D.P.: Potential impacts of a warming climate on water availability in snow-dominated regions, *Nature*, 438, 303—309, <https://doi.org/10.1038/nature04141>, 2005.

Bartelt, P. and Lehning, M.: A physical SNOWPACK model for the Swiss avalanche warning: Part I: Numerical model, *Cold Reg. Sci. Technol.*, 35, 123—145, [https://doi.org/10.1016/S0165-232X\(02\)00074-5](https://doi.org/10.1016/S0165-232X(02)00074-5), 2002.

Baumgartner, F., Jezek, K., Forster, R.R., Gogineni, S.P., and Zabel, I.H.H.: Spectral and angular ground-based radar backscatter measurement of Greenland snow facies, *IEEE Int. Geosci. Remote Sens. Symposium*, 2, 1053—1055, <https://doi.org/10.1109/igarss.1999.774530>, 1999.

Besic, N., Vasile, G., Chanussot, J., Stankovic, S., Dedieu, J.-P., d'Urso, G., Boldo, D., and Ovarlez, J.-P.: Dry snow backscattering sensitivity on density change for SWE estimation, *IEEE Int. Geosci. Remote Sens. Symposium*, 4418—4421, <https://doi.org/10.1109/IGARSS.2012.6350393>, 2012.

Brangers, I., Marshall, H.-P., De Lannoy, G., Dunmire, D., Mätzler, C., and Lievens, H.: Tower-based C-band radar measurements of an alpine snowpack, *The Cryosphere*, 18, 3177—3193, <https://doi.org/10.5194/tc-18-3177-2024>, 2024.

Breiman, L.: Random Forests, *Mach. Learn.*, 45, 5—32, <https://doi.org/10.1023/A:1010933404324>, 2001.

Bühler, Y., Stoffel, A., and Salzmann, C.M.: Photogrammetric Drone Data Dorfberg, EnviDat [data set], <https://www.doi.org/10.16904/envidat.376>, 2022.

Chang, W., Tan, S., Lemmetyinen, J., Tsang, L., Xu, X., and Yueh, S.H.: Dense media radiative transfer applied to SnowScat and SnowSAR, *IEEE J. Sel. Top. Appl. Earth Obs. Remote Sens.*, 7, 3811—3825, <https://doi.org/10.1109/JSTARS.2014.2343519>, 2014.

Colbeck, S.C.: An overview of seasonal snow metamorphism. *Rev. Geophys.*, 20, 45—61, <https://doi.org/10.1029/RG020i001p00045>, 1982.

Copernicus Browser: [https://browser.dataspace.copernicus.eu/?zoom=5&lat=50.16282&lng=20.78613&themeId=DEFAULT-THEME&visualizationUrl=U2FsdGVkX1%2FUoSuL6eKzKziKrUW9wityFi8H7G%2BVZd6v4xK%2Bv4bFpx%2Fn4NoTSqx8d%2BvEQK%2FwZFy93iNrEO1qFZh%2BikB1Pxd12Nbefbz32zhK1kvVuQSktnA1yDFstfz&datasetId=S2\\_L2A\\_CDAS&demSource3D=%22MAPZEN%22&cloudCoverage=30&dateMode=SINGLE](https://browser.dataspace.copernicus.eu/?zoom=5&lat=50.16282&lng=20.78613&themeId=DEFAULT-THEME&visualizationUrl=U2FsdGVkX1%2FUoSuL6eKzKziKrUW9wityFi8H7G%2BVZd6v4xK%2Bv4bFpx%2Fn4NoTSqx8d%2BvEQK%2FwZFy93iNrEO1qFZh%2BikB1Pxd12Nbefbz32zhK1kvVuQSktnA1yDFstfz&datasetId=S2_L2A_CDAS&demSource3D=%22MAPZEN%22&cloudCoverage=30&dateMode=SINGLE), last access: 13 January 2025.

Das, D. and Pandey, A.: Soil moisture retrieval from dual-polarized Sentinel-1 SAR data over agricultural regions using a water cloud model, *Environ. Monit. Assess.*, 197, 52, <https://doi.org/10.1007/s10661-024-13510-4>, 2024.

Dramis, F. and Guglielmin, M.: Permafrost Investigations in the Italian Mountains: The State of the Art, Springer, NATO Sci. S., 76, [https://doi.org/10.1007/978-94-010-0684-2\\_17](https://doi.org/10.1007/978-94-010-0684-2_17), 2001.

Dunmire, D., Lievens, H., Boeykens, L., and De Lannoy, G.J.M.: A machine learning approach for estimating snow depth across the European Alps from Sentinel-1 imagery, *Remote Sens. Environ.*, 314, 114369, <https://doi.org/10.1016/j.rse.2024.114369>, 2024.

EAWS, European Avalanche Warning Service: <https://www.avalanches.org/fatalities/>, last access: 13 January 2025.

Engeset, R.V., Ekker, R., Humstad, T., and LandrÃ, M.: Varsom:regobs - a common real-time picture of the hazard situation shared by mobile information technology, *Proceedings of the International Snow Science Workshop*, Innsbruck, Austria, 1573—1577, <https://arc.lib.montana.edu/snow-science/item/2822>, 2018.

European Union, Copernicus Global Digital Elevation Model (GLO-30): <https://dataspace.copernicus.eu/explore-data/data-collections/copernicus-contributing-missions/collections-description/COP-DEM>, last access: 13 January 2025.

Feng, T., Huang, C., Huang, G., Shao, D., and Hao, X.: Estimating snow depth based on dual polarimetric radar index from Sentinel-1 GRD data: A case study in the Scandinavian Mountains, *Int. J. Appl. Earth Obs.*, 130, 103873, <https://doi.org/10.1016/j.jag.2024.103873>, 2024.

Hauke, J. and Kossowski, T.: Comparison of values of Pearson's and Spearman's correlation coefficients on the same sets of data, *Quest. Geographicae*, 30, 87–93, <https://doi.org/10.2478/v10117-011-0021-1>, 2011.

Jans, J.F., Beernaert, E., De Breuck, M., Brangers, I., Dunmire, D., De Lannoy, G., and Lievens, H.: Sensitivity of Sentinel-1 C-band SAR backscatter, polarimetry and interferometry to snow accumulation in the Alps, *Remote Sens. Environ.*, 316, 114477, <https://doi.org/10.1016/j.rse.2024.114477>, 2025.

Kaplan, G., Fine, L., Lukyanov, V., Manivasagam, V.S., Tanny, J., and Rozenstein, O.: Normalizing the local incidence angle in sentinel-1 imagery to improve leaf area index, vegetation height, and crop coefficient estimations, *Land*, 10, 680, <https://doi.org/10.3390/land10070680>, 2021.

Kapper, K.L., Goelles, T., Muckenhuber, S., Trügler, A., Abermann, J., Schlager, B., Gaisberger, C., Eckerstorfer, M., Grahn, J., Malnes, E., Prokop, A., and Schöner, W.: Automated snow avalanche monitoring for Austria: State of the art and roadmap for future work, *Front. Remote Sens.*, 4, <https://doi.org/10.3389/frsen.2023.1156519>, 2023.

Kendra, J.R., Sarabandi, K., and Ulaby, F.T.: Radar measurements of snow: experiment and analysis, *IEEE Trans. Geosci. Remote Sens.*, 36, 864–879, <https://doi.org/10.1109/36.673679>, 1998.

Keskinen, Z., Hendrikx, J., Eckerstorfer, M., and Birkeland, K.: Satellite detection of snow avalanches using Sentinel-1 in a transitional snow 625 climate. *Cold Reg. Sci. Technol.*, 199, 103558, <https://doi.org/10.1016/j.coldregions.2022.103558>, 2022.

Lemos, A. and Riihelä, A.: Snow depth derived from Sentinel-1 compared to in-situ observations in northern Finland, EGUsphere [preprint], <https://doi.org/10.5194/egusphere-2024-869>, 2024.

Li, H. and Wang, Z. and He, G., and Man, W.: Estimating Snow Depth and Snow Water Equivalence Using Repeat-Pass Interferometric SAR in the Northern Piedmont Region of the Tianshan Mountains, *Journal of Sensors*, 8739598, <https://doi.org/10.1155/2017/8739598>, 2017.

Lievens, H., Demuzere, M., Marshall, H.P., Reichle, R.H., Brucker, L., Brangers, I., de Rosnay, P., Dumont, M., Girotto, M., Immerzeel, W. W., Jonas, T., Kim, E.J., Koch, I., Marty, C., Saloranta, T., Schöber, J., and De Lannoy, G.J.M.: Snow depth variability in the Northern Hemisphere mountains observed from space, *Nat. Commun.*, 10, 4629, <https://doi.org/10.1038/s41467-019-12566-y>, 2019.

Lievens, H., Brangers, I., Marshall, H. P., Jonas, T., Olefs, M., and De Lannoy, G.: Sentinel-1 snow depth retrieval at sub-kilometer resolution 635 over the European Alps, *The Cryosphere*, 16, 159—177, <https://doi.org/10.5194/tc-16-159-2022>, 2022.

Mandal, D., Kumar, V., Ratha, D., Dey, S., Bhattacharya, A., Lopez-Sanchez, J.M., McNairn, H., and Rao, Y.S.: Dual polarimetric radar 640 vegetation index for crop growth monitoring using sentinel-1 SAR data, *Remote Sens. Environ.*, 247, 111954, <https://doi.org/10.1016/j.rse.2020.111954>, 2020.

Mariani, A., Abrahamsen, A.B., Bridle, D., Ingeman-Nielsen, T., Cicora, A., Monti, F., and Marcer, M.: Snowpack and avalanche characterization over the 2021–2022 winter season in Sisimiut, West Greenland, *Front. Earth. Sci.*, 11, 1134728, <https://doi.org/10.3389/feart.2023.1134728>, 2023.

Mariani, A., Dellarole, L., Borsotti, J., Villa, G., and Monti, F.: A new operational approach for the daily assessment of potential avalanche danger, *Proceedings of the International Snow Science Workshop*, Tromsø, Norway, 23–29 September 2024, 116–123, <https://arc.lib.montana.edu/snow-science/item.php?id=3122>, 2024.

Mätzler, C.: Applications of the interaction of microwaves with the natural snow cover, *Remote Sensing Reviews*, 2, 259–387, <https://doi.org/10.1080/02757258709532086>, 1987.

McClung, D. and Schaefer, P. A.: The avalanche handbook, fourth ed., The Mountaineers Books, Seattle, USA, <https://www.mountaineers.org/books/books/the-avalanche-handbook-4th-edition>, 2023.

Mitterer, C., Techel, F., Fierz, C., and Schweizer, J.: An operational supporting tool for assessing wet-snow avalanche danger, Proceedings of the International Snow Science Workshop, Grenoble, Chamonix Mont-Blanc, 334—338, <https://arc.lib.montana.edu/snow-science/item.php?id=1860>, 2013.

Monti, F., Steinkogler, W., and Mitterer, C.: Livigno (Italy) Freeride project, Proceedings of the International Snow Science Workshop, Banff, Canada, 1066—1070, <https://arc.lib.montana.edu/snow-science/item.php?id=2201>, 2014.

Monti, F., Mitterer, C., Steinkogler, W., Bavay, M., and Pozzi, A.: Combining snowpack models and observations for better avalanche danger assessments, Proceedings of the International Snow Science Workshop, Breckenridge, CO, USA, 343—348, <https://arc.lib.montana.edu/snow-science/item.php?id=2290>, 2016.

Nagler, T. and Rott, H.: Retrieval of Wet Snow by Means of Multitemporal SAR Data, *IEEE T. Geosci. Remote*, 38, 754–765, doi: 10.1109/36.842004, 2000.

Nagler, T., Rott, H., Ripper, E., Bippus, G., and Hetzenegger, M.: Advancements for snowmelt monitoring by means of Sentinel-1 SAR, *Remote Sens.*, 8, 348, <https://doi.org/10.3390/rs8040348>, 2016.

Oveisgharan, S., Zinke, R., Hoppinen, Z., and Marshall, H. P.: Snow water equivalent retrieval over Idaho – Part 1: Using Sentinel-1 repeat-pass interferometry, *The Cryosphere*, 18, 559–574, <https://doi.org/10.5194/tc-18-559-2024>, 2024.

Paloscia, S., Pettinato, S., Santi, E., and Valt, M.: Cosmo-skymed image investigation of snow features in alpine environment, *Sensors*, 17, 84, <https://doi.org/10.3390/s17010084>, 2017.

Patil, A., Singh, G., and Rüdiger, C.: Retrieval of snow depth and snow water equivalent using dual polarization SAR data, *Remote Sens.*, 665 12, 1183, <https://doi.org/10.3390/rs12071183>, 2020.

Pettinato, S., Santi, E., Brogioni, M., Paloscia, S., Palchetti, E., and Xiong, C.: The potential of COSMO-SkyMed SAR images in monitoring snow cover characteristics, *IEEE Geosci. Remote S.*, 10, 9—13, <https://doi.org/10.1109/LGRS.2012.2189752>, 2013.

Picard, G., Leduc-Leballeur, M., Banwell, A.F., Brucker, L., and Macelloni, G.: The sensitivity of satellite microwave observations to liquid water in the Antarctic snowpack, *The Cryosphere*, 16, 5061–5083. <https://doi.org/10.5194/tc-16-5061-2022>, 2022.

Plattner, C.H., Braun, L.N., and Brenning, A.: Spatial variability of snow accumulation on Vernagtferner, Austrian Alps, in winter 2003/2004, *Z. Gletscherkunde Glazialgeologie*, 39, 43—57, [https://www.academia.edu/825084/Plattner\\_C\\_L\\_N\\_Braun\\_and\\_A\\_Brenning\\_2004\\_The\\_spatial\\_variability\\_of\\_snow\\_accumulation\\_on\\_Vernagtferner\\_Austrian\\_Alps\\_in\\_Winter\\_2003\\_2004\\_Zeitschrift\\_f%C3%BCr\\_Gletscherkunde\\_und\\_Glazialgeologie\\_39\\_43\\_57](https://www.academia.edu/825084/Plattner_C_L_N_Braun_and_A_Brenning_2004_The_spatial_variability_of_snow_accumulation_on_Vernagtferner_Austrian_Alps_in_Winter_2003_2004_Zeitschrift_f%C3%BCr_Gletscherkunde_und_Glazialgeologie_39_43_57), 2004.

Reppucci, A., Banque, X., Zhan, Y., Alonso, A., and López-Martinez, C.: Estimation of snow pack characteristics by means of polarimetric SAR data, *Remote Sens. Agr. Ecosyst. Hydrol. XIV*, 85310Z, <https://doi.org/10.1117/12.974598>, 2012.

Rott, H., Nagler, T., and Scheiber, R.: Snow mass retrieval by means of SAR interferometry, *Proc. of FRINGE Workshop*, Frascati, Italy, 1—5, [https://www.researchgate.net/publication/228990055\\_Snow\\_mass\\_retrieval\\_by\\_means\\_of\\_SAR\\_interferometry](https://www.researchgate.net/publication/228990055_Snow_mass_retrieval_by_means_of_SAR_interferometry), 2003.

Rott, H., Cline, D.W., Duguay, C., Essery, R., Etchevers, P., Hajnsek, I., Kern, M., MacElloni, G., Malnes, E., Pulliainen, J., and Yueh, S.H.: CoReH2O, a dual frequency radar mission for snow and ice observations, *IEEE Int. Geosci. Remote S.*, 5550—5553, <https://doi.org/10.1109/IGARSS.2012.6352348>, 2012.


Rott, H., Scheiblauer, S., Wuite, J., Krieger, L., Floricioiu, D., Rizzoli, P., Libert, L., and Nagler, T.: Penetration of interferometric radar signals in Antarctic snow, *The Cryosphere*, 15, 4399—4419, <https://doi.org/10.5194/tc-15-4399-2021>, 2021.

Small, D.: Flattening gamma: Radiometric terrain correction for SAR imagery, *IEEE Trans. Geosci. Remote S.*, 49, 3081—3093, <https://doi.org/10.1109/TGRS.2011.2120616>, 2011.

Snehmani, Singh, M.K., Gupta, R.D., Bhardwaj, A., and Joshi, P. K.: Remote sensing of mountain snow using active microwave sensors: a review, *Geocarto Int.*, 30, 1—27, <https://doi.org/10.1080/10106049.2014.883434>, 2015.

Soncini, A. and Bocchiola, D.: Assessment of future snowfall regimes within the Italian Alps using general circulation models, *Cold Reg. Sci. Technol.*, 68, 113—123, <https://doi.org/10.1016/J.COLDREGIONS.2011.06.011>, 2011.

Strozzi, T., A. Wiesmann, and C. Mätzler: Active microwave signatures of snow covers at 5.3 and 35 GHz, *Radio Sci.*, 32, 479—495. <https://doi.org/10.1029/96RS03777>, 1997.

Tiuri, M., Sihvola, A., Nyfors, E., and Hallikainen, M.: The complex dielectric constant of snow at microwave frequencies, *IEEE J. Oceanic Eng.*, 9, 377—382. <https://doi.org/10.1109/JOE.1984.1145645>, 1984.

Tompkin, C. and Leinss, S.: Backscatter Characteristics of Snow Avalanches for Mapping With Local Resolution Weighting, *IEEE J. Sel. Top. Appl. Earth Obs. Remote Sens.*, 14, 4452—4464, <https://doi.org/10.1109/JSTARS.2021.3074418>, 2021.

Tsai, Y.L.S., Dietz, A., Oppelt, N., and Kuenzer, C.: Remote sensing of snow cover using spaceborne SAR: A review, *Remote Sens.*, 11, 1456, <https://doi.org/10.3390/rs11121456>, 2019.

Tsang, L., Durand, M., Derksen, C., Barros, A.P., Kang, D.H., Lievens, H., Marshall, H.P., Zhu, J., Johnson, J., King, J., Lemmetyinen, J., Sandells, M., Rutter, N., Siqueira, P., Nolin, A., Osmanoglu, B., Vuyovich, C., Kim, E., Taylor, D., Merkouriadi, I., Brucker, L., Navari, M., Dumont, M., Kelly, R., Kim, R.S., Liao, T.H., Borah, F., and Xu, X.: Review article: Global monitoring of snow water equivalent using high-frequency radar remote sensing, *The Cryosphere*, 16, 3531—3573, <https://doi.org/10.5194/tc-16-3531-2022>, 2022.

Ulaby, F.T. and Herschel Stiles, W.: Microwave response of snow, *Adv. Space Res.*, 1, 131—149. [https://doi.org/10.1016/0273-1177\(81\)90389-6](https://doi.org/10.1016/0273-1177(81)90389-6), 1981.

Varsom Regobs, Snow observations: <https://regobs.no/?SelectedNumberOfDays=3&&NWLat=72.47527631092942&NWLon=-21.62109375000004&SELat=55.178867663282006&SELon=89.384765625>, last access: 13 January 2025.

Velsand, P.: Comparison and classification of an Arctic Transitional snow climate in Tromsø, Norway, *The Artic University of Norway, Tromsø, Norway*, <https://api.semanticscholar.org/CorpusID:134889729>, 2017.

Wiesmann, A., Strozzi, T., Werner, C., Wegmuller, U., and Santoro, M.: Microwave remote sensing of alpine snow, *IEEE International Geoscience and Remote Sensing Symposium*, 1223—1227, <https://doi.org/10.1109/IGARSS.2007.4423026>, 2007.

Yommy, A. S., Liu, R., and Wu, A. S.: SAR image despeckling using refined Lee filter, *7th Int. C. Intel. Hum. Mach.*, Hangzhou, China, 710 260—265, <https://doi.org/10.1109/IHMSC.2015.236>, 2015.