# Peer review of "Influence of snowpack properties and local incidence angle on SAR signal depolarization: a mathematical model for high-resolution snow depth estimation"

_EGUsphere, 2025_

## Referee Comment (RC1)

**General comments:**

I enjoyed reading this paper, very interesting, relevant research. I was very surprised to see a 10 m resolution backscatter product with such good performance measures. In my opinion a groundbreaking paper that shows the potential of S-1 SAR snow depth retrieval in mountain areas. Some adjustments are however necessary in my opinion to improve the quality even further.

1. Would it be an idea to validate over an area where LIDAR or photogrammetric data is available? NSIDC has freely available ASO L4 Lidar snow depth maps where you can validate against. Also, Envidat has photogrammetric snow depth data available over the Alps (Yves Buhler will be able to help you further). Since the presented product has a very high resolution it would be very interesting to see how its performance holds in such a context. The validation presented now is based on a very small amount of datapoints.

2. Since the high amount of snow depth data available in the Alps maybe it would be better to select in-situ stations located in a range of incidence angles instead of assuming the snow depth being equal in a 4 km area around your in-situ station to extract your local incidence angles. In this context the photogrammetric and LIDAR datasets could also help. I think this will lead to a scientifically more robust relationship between LIA and the SIsar-index considering the use of 10 m backscatter data and the strong spatial variability of snow.

3. Be careful when mentioning the use of dual-polarimetric data, since it can be easily confused with actual PolSAR data. Dual-pol polarimetric variables are derived from the Single-Look Complex (SLC) data, are then processed into covariance matrices to in the end extract metrics like the DpRVI. In this paper only backscatter, and no phase information, is used. I would emphasize this more throughout the text. It would also be interesting to analyze metrics derived from the SLC observations, however I understand this is out of scope.

4. To strengthen the claims made within this paper I would also analyze the behavior of the VV and VH backscatter with changes in snow depth or changes in incidence angle. This will allow the reader to understand the underlying processes related to a change in SIsar. So, in other words, it would be of interest to add an analysis that investigates the change of the components that make up SIsar-index to see exactly where the change in SIsar is coming from.

5. I would in general mention the resolution of the product more, because sometimes I was confused whether 10 or 50 m was used (see specific comments).

6. I sometimes found it hard to follow the flow of the data and methods section. I would opt for a combination of both to improve flow. My suggestion would be to start with the study area, then go to the description of the SAR data and data processing, then followed by the weather and snowpack measurements and snowpack modelling, then the mathematical modelling to combine them both and then the model validation strategy.

**Specific comments:**

Line 36 and 37: I would change SAR signal by microwaves.

Line 44: This statement is not true for cross-pol. I would specify co-pol here.

Line 64: I would be careful with this statement because in the mentioned paper only backscatter was analyzed but no true dual-polarimetric indices (those derived from SLC-data).

Line 70: I would cite Feng et al. again here after Alps.

Line 108: What is the reason for working with a buffer zone of 50 m? Your backscatter data is processed at 10 m, is it not better to take the same resolution?

Line 115: independent as in your pixel AWS pixel was not taken into account for the LIA sampling? Since I saw your AWS pixel and LIA area overlap. Also, what is the resolution used for the LIA sampling is it 50 m, or 10 m?

Line 117: idem, why 50 m? Also, how many measurements were taken as validation points?

Line 127: 168 is descending captured in the morning, 15 is ascending captured in the evening . Idem on line 135 descending morning and ascending evening.

Line 133: Mention the source of the downloads.

Line 135: Mention why these dates are used.

Figure 1: I would add the validation area as well.

Line 169: What was the original resolution of your GRD product, is this 10 by 10 m? Why is resampling needed in line 175 or was it a reprojection to another CRS? If resampled, which technique was used?

Line 185: Indicate the resolution, also in section 3.4 indicate whether you also do an upscaling to 50 m. If this is the case it is maybe better to immediately indicate in section 3.2 that you are using a 50 m resolution to avoid confusion.

Line 190: Indicate why you would exclude negative values and why you would make sure the data follow a normal distribution.

Line 205: Explain what the acronym mdl stands for the first time it is used, same for msr.

Line 225: Is this one $a$ value for each orbit and regression area or is this everything combined in one $a$ value?

Line 233-235: This is a very tricky statement to make which I would not necessarily agree with. See general comments 2.

Line 243: Did you exclude values from your validation set? Or is this sentence related to the calibration? If it is related to the calibration, I would leave this sentence out because it is a bit confusing, otherwise there is a mistake here since excluding data from your validation set because they have negative values is not good. It will lead to performance metrics that are unrealistic since you have tuned it to artificially create better results.

Line 266: Would it be an option to set an extra parameter that takes care of the slope? Why does it necessarily have to be 0?

Line 268: I would add the sentences on line 315-316 here to immediately bring clarity on why you only take one variable.

Line 302: errors drop significantly, did you do a statistical analysis?

Line 307: I do not think I understand this well, what does c1 * g_bar explain? You could make it equal to $a$ if c1 would be lower but then you would just have a worse correlation between HSmsr and HSmdl_LIA. I do not see the added value of this sentence.

Line 321: larger snow mass and more grains per unit volume of snow.

Line 327: Can this be related to signal attenuation? I think I would mention it here.

Line 332: Is this not because your SIsar index is also dependent on soil backscatter processes?

Line 321-335: This is quite speculative since your index depends on both VV and VH but you only describe HH here, what about the change of VH during those periods of GS increase and density increase?

Line 360-362: I agree but I do not think this is relevant here since your SIsar measurement is a measurement over time and this sentence only explains what happens at one point in time. It is the change in VV under a certain LIA that is critical.

Line 366: The statement that VH backscatter increases is not correct. Depolarization will increase but only because VV backscatter will decrease more than VH backscatter if you go to higher local incidence angles. The specular reflection captured by the sensor will decrease leading to a decrease in both VV and VH (but a stronger decrease in VV than VH). See figure below for the effect of LIA on the volume scattering component of a snowpack (simulated using a radiative transfer model DMRT-Bic on a snowpack of 2 and 4 m):

[Figure]

[Figure]

Figure 7: The change in backscatter from one LIA to the other is not well represented here, change the magnitudes of VH and VV accordingly. The depolarization depends on the ratio of the two. (see previous comment and see comment 4 in the general comments section).

Line 376: Is this a valid approach if HS is considered equal over the entire study area where LIAs were extracted? See comment 2 in the general comments section.

Line 414: Is this a hypothesis or did you see this in the data, if you saw it in the data, please include it, very interesting stuff.

**Technical corrections:**

Line 19: Set snowpack to snow.

Line 23: I would rephrase to snow monitoring is critically important for avalanche forecasting

Line 78: sentence difficult to read, I would rephrase.

Line 123: are available and is right-looking

Line 233: derived instead of deriving

Line 423: high number of pixels

---

## Author Comment (AC1)

July 1, 2025

Reply to Referee 1

I enjoyed reading this paper, very interesting, relevant research. I was very surprised to see a 10 m resolution backscatter product with such good performance measures. In my opinion a groundbreaking paper that shows the potential of S-1 SAR snow depth retrieval in mountain areas. Some adjustments are however necessary in my opinion to improve the quality even further.

Dear Jonas-Frederik Jans

Thank you for your detailed and constructive comments on our manuscript. In addition to improving the clarity of our article, they allowed us to obtain other interesting results. Please find below our replies (in blue) describing how we will address your comments in the revised manuscript.

**General comments:**

1. Would it be an idea to validate over an area where LIDAR or photogrammetric data is available? NSIDC has freely available ASO L4 Lidar snow depth maps where you can validate against. Also, Envidat has photogrammetric snow depth data available over the Alps (Yves Buhler will be able to help you further). Since the presented product has a very high resolution it would be very interesting to see how its performance holds in such a context. The validation presented now is based on a very small amount of datapoints.

   We initially considered the possibility of using the dataset you mentioned however those data were mainly collected in March or April and the Nagler's method detected the presence of wet snow. We agree that the only limitation of our approach relies in the fact that we did not consider a large amount of data, for this reason we managed to obtain an additional dataset consisting of photogrammetric data to validate our model. Such dataset was not available to us when we submitted the first version of the paper, at the end of this document we present the new methods and results. Anyway, we would like to highlight that, in contraposition to most papers present in the literature, our work focuses on high quality measurements to be able to precisely determine the effects of the snowpack variable and of the LIA on SAR signal depolarization. In our opinion this ensures that our manuscript represents a novelty in the literature.

2. Since the high amount of snow depth data available in the Alps maybe it would be better to select in-situ stations located in a range of incidence angles instead of assuming the snow depth being equal in a 4 km area around your in-situ station to extract your local incidence angles. In this context the photogrammetric and LIDAR datasets could also help. I think this will lead to a scientifically more robust relationship between LIA and the SIsar-index considering the use of 10 m backscatter data and the strong spatial variability of snow.

   As explained in the previous reply, we will add a validation carried out using photogrammetric data. Concerning the automatic weather stations for the calibration of our model, it is not possible to do what you suggest.

Firstly, AWSs are always located on flat terrain (in Italy and Switzerland this is a mandatory requirement), and so we realized that the range of LIA we could cover was very limited and bad distributed. Moreover, we remark that very often automatic weather stations are not well-calibrated, and we cannot perform reliable SNOWPACK simulations, while, on the other hand, the ones considered by us are checked every week by the Alpsolut team. Finally, we think that the relation between the SIsar index and the LIA derived is robust (see e.g. Figure 3) thanks to the fact that we were able to consider very precise measurements to validate our results. For example, the $R^2$ coefficient resulting from the derivation of $g(LIA)$ is equal to 0.695 (we will add this result in the revised version of the manuscript). The reliability of those measurements is demonstrated by the fact that they were used for several papers during the past years, see for example: (Monti et al., 2014), (Monti et al., 2016), *Simulation of snow management in Alpine ski resorts using three different snow models* (Hanzer et al., *Cold Regions Science and Technology*, 172, 102995, 2020). Anyway, the results that we will present deriving from the additional validation carried out using photogrammetric data will show that our model is robust, implying that the initial calibration that we made with the Livigno AWSs is valid.

3. Be careful when mentioning the use of dual-polarimetric data, since it can be easily confused with actual PolSAR data. Dual-pol polarimetric variables are derived from the Single-Look Complex (SLC) data, are then processed into covariance matrices to in the end extract metrics like the DpRVI. In this paper only backscatter, and no phase information, is used. I would emphasize this more throughout the text. It would also be interesting to analyze metrics derived from the SLC observations, however I understand this is out of scope.

   We will explain this better in the revised version of the manuscript. For example, we will add the following sentence in the Introduction: *The $DpRVI_c$ index is a simplified version of the $DpRVI$ index, adapted to be computed from a Ground Range Detected (GRD) Sentinel-1 product (Feng et al., 2024).* We agree that analyzing the metrics derived from the SLC observations would be interesting, however, since it is out of scope, we will not do so.

4. To strengthen the claims made within this paper I would also analyze the behavior of the VV and VH backscatter with changes in snow depth or changes in incidence angle. This will allow the reader to understand the underlying processes related to a change in SIsar. So, in other words, it would be of interest to add an analysis that investigates the change of the components that make up SIsar-index to see exactly where the change in SIsar is coming from.

   We will add an evalutaion of the single components of our index (VV and VH backscatter coefficients) with respect to LIA in order to demonstrate our conclusions. See reply to Line 366.

5. I would in general mention the resolution of the product more, because sometimes I was confused whether 10 or 50 m was used (see specific comments).

   In the revised version of the manuscript we will mention the resolution of the product more. See the replies under the specific comments (Line 108;

 for a detailed description of the changes we will make.

6. I sometimes found it hard to follow the flow of the data and methods section. I would opt for a combination of both to improve flow. My suggestion would be to start with the study area, then go to the description of the SAR data and data processing, then followed by the weather and snowpack measurements and snowpack modelling, then the mathematical modelling to combine them both and then the model validation strategy.

   *We do not agree on this. We would prefer to keep the sections separate because we believe they help to better present the topics covered. We would like to avoid writing sections that are too long and contain too much information of partially different nature.*

**Specific comments:**

Line 36 and 37: I would change SAR signal by microwaves.
*We will change the wording as suggested.*
Line 44: This statement is not true for cross-pol. I would specify co-pol here.
*We will specify co-pol as suggested.*
Line 64: I would be careful with this statement because in the mentioned paper only backscatter was analyzed but no true dual-polarimetric indices (those derived from SLC-data).
*We will specify better that DpRVIc and SIsar are just approximations of the real depolarization. However these approximations seems to be considered reliable in other works we cited.*
Line 70: I would cite Feng et al. again here after Alps.
*We will add this citation.*
Line 108: What is the reason for working with a buffer zone of 50 m? Your backscatter data is processed at 10 m, is it not better to take the same resolution?
*We considered $50 \times 50$ m areas, hence a 25 m square buffer (there was a misprint in the original paper). The pixels had a resolution of $10 \times 10$ m (standard GRD resolution), but considering the average value was the optimal choice to balance spatial resolution and temporal signal stability (similarly to what was done by Pettinato et al.(2014)). We will modify the final paragraph of Section 3.2 in the following way: The SIsar values in the time series were computed by averaging the values within the $50 \times 50$ m regression sampling areas for the two different AGs (similarly to Pettinato et al.(2014), we noticed that this was the optimal choice to balance spatial resolution and temporal signal stability). [...] For analyzing the dependence between our index and the LIA, we grouped and averaged the SIsar values for LIA classes of $2°$ within the two LIA sampling areas (in this case, we considered the values of each $10 \times 10$ m pixel separately and no averaging was performed since each pixel was characterized by a different LIA). [...] The SIsar values related to the $50 \times 50$ m validation sampling areas were again computed by averaging the values of each pixel.*
Line 115: independent as in your pixel AWS pixel was not taken into account for the LIA sampling? Since I saw your AWS pixel and LIA area overlap. Also, what is the resolution used for the LIA sampling is it 50 m, or 10 m?
*You are correct, they are not perfectly independent. We will substitute that sentence with: It is important to note that the regressions sampling areas were included within the LIA sampling areas, but the former represented only a small*

*fraction of the latter. Therefore, despite the partial overlap, the results obtained from analyzing the second area were largely independent from those related to the first, as they were mainly driven by new additional data.*

Similarly, we will correct Lines 248 and 385 by substituting *independent* with *effectively independent* (or a similar expression).

The resolution of the LIA sampling area was 10 m, in order to be able to analyze each pixel separately, so that we could understand the influence of the LIA (each pixel is characterized by a different LIA, hence we did not perform any averaging process; see reply to Line 108).

Line 117: idem, why 50 m? Also, how many measurements were taken as validation points?

See reply to Line 108. The number of observation is reported at line 151.

Line 127: 168 is descending captured in the morning, 15 is ascending captured in the evening . Idem on line 135 descending morning and ascending evening.

We will fix this error in the revised version of the manuscript.

Line 133: Mention the source of the downloads.

We will mention the source of the downloads.

Line 135: Mention why these dates are used.

We used these dates because in Tromso there was a snow expert who made several snow measurements and he was available at verifying our results (see Lines 116 and 150). For example, he confirmed that the snow was dry and no avalanche deposits were present in the area (see Line 151). We will add his name in the acknowledgments of the revised version of the manuscript.

Figure 1: I would add the validation area as well.

We prefer not to do this for three reasons. Firstly, a map of the calibration area is very important in Section 2.1 because it was chosen by us to calibrate the model, while, on the other hand, the validation area was a forced choice to incorporate the field measurements. Secondly, Figure 6 already contains a map of the validation area, with the simulated height of the snowpack. Finally, since the two areas are so far apart we do not like the idea of including them both in the same image. However, if our additional validation (see end of this document) is accepted, we will have to add a map of the new validation area. In that case, we will also add the map of this validation area to that image, so that there are two images: one dedicated to the calibration areas and one to the validation areas.

Line 169: What was the original resolution of your GRD product, is this 10 by 10 m? Why is resampling needed in line 175 or was it a reprojection to another CRS? If resampled, which technique was used?

The resolution was 10 by 10 m, we will clarify this at the beginning of Section 3.2. The resampling part was a misprint, we will remove it from the revised version of the manuscript.

Line 185: Indicate the resolution, also in section 3.4 indicate whether you also do an upscaling to 50 m. If this is the case it is maybe better to immediately indicate in section 3.2 that you are using a 50 m resolution to avoid confusion.

As explained in the replies to Lines 108 and 115, we will explain better in Section 3.2 the resolution we used.

Line 190: Indicate why you would exclude negative values and why you would make sure the data follow a normal distribution.

Concerning the normal distribution, it is not relevant, we will remove it from the revised version of the manuscript. On the other hand, we understand that we did not explain well how we dealt with negative $SIsar$ values, here we report

the changes that we will make. We will modify the last paragraph of Section 3.2 as follows (see also reply to Line 108):

*The SIsar values in the time series were computed by averaging the values within the $50 \times 50$ m regression sampling areas for the two different AGs (similarly to Pettianto et al.(2014), we noticed that this was the optimal choice to balance spatial resolution and temporal signal stability). The pixels with negative SIsar values along the snow season were excluded and we hypothesized that this anomalous effect was related to the LIA. For analyzing the dependence between our index and the LIA, we grouped and averaged the SIsar values for LIA classes of $2°$ within the two LIA sampling areas (in this case, we considered the values of each $10 \times 10$ m pixel separately and no averaging was performed since each pixel was characterized by a different LIA). This last analysis was conducted under snow-free conditions as well as on several dates during the snow season. In this case, the pixels with negative SIsar values were not excluded in order to verify our previous hypothesis. The SIsar values related to the $50 \times 50$ m validation sampling areas were again computed by averaging the values of each pixel (in this case we did not remove the pixels with negative SIsar values; the reason of this different choice will be clear in Section 4.2).*

Hence, we will also modify Section 4.2 as follows:

*Since we observe that for LIA values below approximately $30°$ the SIsar index is always negative, even when HS is nonzero, we exclude all areas with $LIA < 30°$ in the implementation of the final model (this confirms our initial hypothesis that those negative SIsar values are related to the LIA; see Section 3.2). Areas with $LIA > 80°$ are also removed because they are more prone to SAR shadowing errors. The resulting $R^2$ coefficient is equal to $0.695$ and, due to these choices, Eq. (9) is well-defined since $g(LIA) \neq 0$ for all LIAs of interest.*

Line 205: Explain what the acronym mdl stands for the first time it is used, same for msr.

We will explain this in the revised version of the manuscript: mdl stands for modelled, while msr for measured.

Line 225: Is this one $a$ value for each orbit and regression area or is this everything combined in one $a$ value?

The value of $a$ was derived combining all the data. By doing so we obtained a more reliable value that should be independent on the date and the location. Figure 5(a) shows the corresponding graph, highlighting that all the data were used. Moreover, in Section 3.2 is explained that both regression sampling areas and satellite orbits were used.

Line 233-235: This is a very tricky statement to make which I would not necessarily agree with. See general comments 2.

We understand that it could be seen as a forced assumption but we ensure you that it is not. The great advantage of the areas we selected for the analysis of the LIA and of the influence of the snowpack properties is that they are close to the Alpsolut avalanche center and hence carefully monitored by several (about five) avalanche forecasters (some of them are among the authors of our manuscript). As explained in the paper (Section 2.3), the avalanche forecasters of the Alpsolut avalanche center carry out several snow measurements every week in different locations and the LIA sampling areas were defined taking into account their reports.

Line 243: Did you exclude values from your validation set? Or is this sentence related to the calibration? If it is related to the calibration, I would leave this sentence out because it is a bit confusing, otherwise there is a mistake here since excluding data from your validation set because they have negative values is not good. It will lead to performance metrics that are unrealistic since you

have tuned it to artificially create better results.

We excluded values from the calibration set. We agree that we should remove this sentence: not only it is confusing but we will provide more details regarding how we dealt with negative $SIsar$ values in the previous sections (see reply to Line 190). Obviously we did not manipulate the data to obtain better results. We also modified the second paragraph of the appendix as follows (recall the reply to Line 190):

*Furthermore, we assessed the model's capability to distinguish between snow-free and snow-covered conditions. In Fig. A1 are shown the values of $HS_{mdl\_LIA}$ during the summer 2023 computed for the Gessi regression sampling area. During the observed summer period, the RMSE was $30.5$ cm and the MAE $25.6$ cm, consistent with snow-covered seasons. Negative $SIsar$ values were common and expected, as even small model errors can yield negative estimates when snow depth is zero; this behavior is not related to the $LIA$ (see Sections 3.2 and 4.2). Due to the modelled HS fluctuations, it is not possible to detect snow-free conditions with our model. Anyway, on average $HS_{mdl\_LIA} \simeq 0$, which is in line with the results presented in Section 4.2. Similarly, we noticed the presence of pixels characterized by negative $SIsar$ values under snow-covered conditions when the snowpack is very thin.*

Line 266: Would it be an option to set an extra parameter that takes care of the slope? Why does it necessarily have to be 0?

We think you meant the intercept, not the slope. We decided to set the intercept equal to zero for two reasons. Firstly, we wanted to develop an index satisfying the theoretical property $SIsar = 0$ in absence of snow (see Section 3.3 and Eq. (6)). Therefore we looked for a relation of our index with a snowpack variable satisfying this condition. Obviously, not necessarily such variable exists. However, we demonstrated through a statistical analysis (see Table 1) that the snowpack height satisfies such property and hence we set the intercept equal to zero. Indeed, the p-value showed that the intercept is not significant, implying that considering it would have led to almost equal performances (see Line 268) but also to a weaker model under a theoretical point of view. We also discussed (see Section 5.1) why only for $HS$ the intercept could be set equal to zero, explaining the difficulties of deriving other snowpack information from SAR data with our approach.

Line 268: I would add the sentences on line 315-316 here to immediately bring clarity on why you only take one variable.

We do not agree on this. In Sections 3.3 and 4.1 we already explained in depth why we only took one variable.

Line 302: errors drop significantly, did you do a statistical analysis?

Yes, the subsequent sentence explains that the errors decrease by about 39 % (we think that clearly this represents a significant reduction in errors).

Line 307: I do not think I understand this well, what does c1 * g_bar explain? You could make it equal to a if c1 would be lower but then you would just have a worse correlation between HSmsr and HSmdl_LIA. I do not see the added value of this sentence.

We agree. We will remove this sentence in the revised version of the manuscript.

Line 321: larger snow mass and more grains per unit volume of snow.

We will change the wording as suggested.

Line 327: Can this be related to signal attenuation? I think I would mention it here.

Absolutely! Thanks for the suggestion, we will mention this fact in the revised version of the manuscript.

Line 332: Is this not because your SIsar index is also dependent on soil backscatter processes?

We do not believe this is a significant issue – or rather, we believe that the soil effect can be neglected. This is because, in our analysis of S1-SAR data, we are focusing on variations every 12 days, and we can reasonably assume that soil properties do not change during the winter season, when the ground is covered by a significant amount of snow.

Line 321-335: This is quite speculative since your index depends on both VV and VH but you only describe HH here, what about the change of VH during those periods of GS increase and density increase?

Yes, the interpretation we propose is indeed speculative. We acknowledge that we are offering a possible explanation, and we believe it is clearly presented as such in the manuscript (see Lines 320, 323, and 327). We have chosen to retain this section because it highlights a potentially valuable line of inquiry that could be of interest to other researchers. In our view, this kind of exploratory reasoning can help stimulate further investigation and discussion within the field.

Line 360-362: I agree but I do not think this is relevant here since your SIsar measurement is a measurement over time and this sentence only explains what happens at one point in time. It is the change in VV under a certain LIA that is critical.

In fact, here we are discussing how the coefficient varies with the LIA at a specific point in time, not how it changes over time. In our opinion, this is important and it's already explained clearly, as we specifically mention "constant HS".

Line 366: The statement that VH backscatter increases is not correct. Depolarization will increase but only because VV backscatter will decrease more than VH backscatter if you go to higher local incidence angles. The specular reflection captured by the sensor will decrease leading to a decrease in both VV and VH (but a stronger decrease in VV than VH). See figure below for the effect of LIA on the volume scattering component of a snowpack (simulated using a radiative transfer model DMRTBic on a snowpack of 2 and 4 m):

We performed the proposed analysis and we will add this within the revised version of the manuscript, in the main sections. Here we will anticipate you the results (see figure below, here named Figure 0).

In Section 4.2, at the end of the second paragraph, we will add : *Figure 0 reports the behaviors of the single backscatter components $\gamma^0_{VV}$ and $\gamma^0_{VH}$ as the LIA varies and under the same conditions of Fig. 3. Note that $\gamma^0_{VV}$ increases with HS for LIAs below approximately $20°$, while it appears to be independent on HS for higher LIAs. However, it decreases as the LIA varies between $20°$ and $80°$. On the other hand, $\gamma^0_{VH}$ shows a more complex behavior as the LIA varies. Interestingly, for LIA values below $30°$, it reaches lower values under snow-covered conditions than under snow-free conditions.*

In Section 5.2 we will write: *The negative values of the SIsar index for low LIAs can be firstly explained by the increase of $\gamma^0_{VV}$ with HS, as reported in Fig. 0(a). This behavior of $\gamma^0_{VV}$ for low LIAs is consistent with the findings of Jans et al. (2025), which highlighted that the Sentinel-1 co-polarized backscatter coefficient ($\sigma^0_{VV}$) increases with HS only for small incidence angles. Secondly, at these low LIAs, Fig. 0(b) shows that $\gamma^0_{VH}$ assumes lower values under snow-covered conditions compared to snow-free conditions, implying a decrease in the $DpRVI_c$ index. [. . .] For LIA values between approximately $30°$ and $50°$, the increase of the SIsar index with the LIA for a constant HS can be explained by the fact that, as the LIA increases, $\gamma^0_{VV}$ significantly decreases (Keskinen et*

al., 2022), while $\gamma_{\mathrm{VH}}^0$ slightly increases in presence of snow (see Fig. 0). Furthermore, in this range of LIAs, $\gamma_{\mathrm{VV}}^0$ is not sensitive to HS, while $\gamma_{\mathrm{VH}}^0$ tends to increase as HS increases because a larger volume of snow is traversed by the SAR signal, leading to more opportunities for signal depolarization and different interactions with layered structures. The fact that $\gamma_{\mathrm{VV}}^0$ shows no dependence on HS indicates that it is primarily influenced by soil properties; this supports a hypothesis proposed by several authors (e.g., Lievens et al. (2019), Dunmire et al. (2024), and Feng et al. (2024)), who suggested that the cross-polarization ratio $\gamma_{\mathrm{VH}}^0/\gamma_{\mathrm{VV}}^0$ or the $DpRVI_c$ index are more effective for HS retrieval than relying on $\gamma_{\mathrm{VH}}^0$ alone, as these indices are designed to reduce the influence of soil, present in both $\gamma_{\mathrm{VV}}^0$ and $\gamma_{\mathrm{VH}}^0$, and isolate the snow contribution.

Note that the behavior of $\gamma_{\mathrm{VV}}^0$ reported in Fig. 0(a) is similar to the one of your numerical simulation. On the other hand, the behavior of $\gamma_{\mathrm{VH}}^0$ is different, however it is in line with our findings.

[Figure]

Figure 7: The change in backscatter from one LIA to the other is not well

[Figure]

represented here, change the magnitudes of VH and VV accordingly. The depolarization depends on the ratio of the two. (see previous comment and see comment 4 in the general comments section).

We agree and we prefer to remove this figure from the revised version of the manuscript for three reasons: it has a lower quality compared to the other images; it is probably not necessary; we will add other images due to the additional results we obtained following your suggestions and we do not want to insert too many figures.

Line 376: Is this a valid approach if HS is considered equal over the entire study area where LIAs were extracted? See comment 2 in the general comments section.

Yes, since $HS$ was constant, by dividing the $SIsar$ index by it we isolated the function $g(LIA)$, which represents the influence of the $LIA$ (see Eq. (8) and the final part of Section 3.3). Recall that to derive the function $g$ we considered only data deriving from the beginning of the snow season, when we could assume that $HS$ was representative of the entire $LIA$ sampling areas (see, for example, the caption of Figure 4 and see also the reply to Line 233–235). The assumption of generally constant $HS$ was also observed by our avalanche experts in that period.

Line 414: Is this a hypothesis or did you see this in the data, if you saw it in the data, please include it, very interesting stuff.

We saw this in the data and we will add this.

**Technical corrections:**

Line 19: Set snowpack to snow.
We will change the wording as suggested.
Line 23: I would rephrase to snow monitoring is critically important for avalanche forecasting
We will change the wording as suggested.
Line 78: sentence difficult to read, I would rephrase.
We will rephrase the sentence as follows: *The aim of this study is to provide a detailed analysis of the capability of the $DpRVI_c$ index to retrieve snow depth and other snowpack properties at a small spatial scale.*
Line 123: are available and is right-looking
We will change the wording as suggested.
Line 233: derived instead of deriving
We will change the wording as suggested.
Line 423: high number of pixels
We will change the wording as suggested.

**Photogrammetric data**

We conclude by presenting an additional validation of our model based on photogrammetric data. In the following, we will present methods, results, and discussion related to this new validation. Obviously, small changes must be made to the entire paper in order to add these new parts and to link them with the rest of the analysis, however here we will not report all of them for brevity and because they are limited to few sentences in abstract, introduction, and conclusions.

*2.1 Study areas*

*The second model validation area is located near Davos, Switzerland (46°49′20″ N 09°50′02″ E) and corresponds to the south-facing slope of Salezerhora peak (2537 m a.s.l.). This region ranges in altitude from 2500 m a.s.l. to 1660 m a.s.l. and is characterized by alpine meadows, with a few small forested sections that have been excluded. [...] Concerning the Davos validation area, we considered an area of approximately 2 km$^2$ were a photogrammetric snow depth product was available.*

We will add an image containing both validation areas (see also reply to Figure 1).

*2.2 SAR data*

*For the Davos validation area, we downloaded Sentinel-1 GRD products acquired on 9 January 2022 at 5 p.m. UTC (AG 15).*

*2.3 Weather and snowpack measurements*

*To the Davos validation area is associated a snow depth raster (as will be shown later in the paper, our analysis focused on snow depth retrieval) obtained by differentiating a summer digital surface model (DSM) realized with a UAV–photogrammetric survey with another DSM carried out on 12 January 2022 in dry snow condition (Bühler et al., 2022). The dataset has an original resolution of 10 cm. These measurements were made three days after the corresponding SAR acquisitions, but the snowpack did not change significantly during that period, so the temporal offset is not expected to strongly affect the comparison.*

*3.2 SAR data processing*

*The SIsar values related to the 50 × 50 m Norwegian validation sampling areas were again computed by averaging the values of each pixel. The Swiss validation area was treated in a similar manner: we computed SIsar values over the SAR scene at 50 m spatial resolution and applied a 3×3 pixel median filter to reduce outliers and to handle for a few missing values in the photogrammetric raster. Subsequently, the photogrammetric snow depth raster was resampled to the same resolution using bilinear interpolation. The presence of dry snow was verified with the Nagler's method in all validation areas, moreover for the validation we did not remove the pixels with negative SIsar values (the reason of this different choice will be clear in Section 4.2).*

*3.4 Model validation strategy*

*Finally, using again Eq. (9), we computed the values of $Xsv_{mdl\_LIA}$ for the Swiss validation area. Those values were compared to the photogrammetric data to validate the model with a large dataset.*

*4.3 Models comparison and validation*

*Finally, using Eq. (9) to estimate HS in the validation area around Davos (the Swiss validation dataset; see Fig. 1) we obtain an RMSE of 22.4 cm and a MAE of 18.1 cm.*

[Figure]

[Figure]

Figure 1: *Davos validation area. (a) Difference between $HS_{mdl\_LIA}$ and $HS_{photogrammetric}$, derived from the photogrammetric survey. (b) $HS_{mdl\_LIA}$ vs. $HS_{photogrammetric}$.*

*5.3 Effectiveness of considering the local incidence angle within the model*

*The quality of snow depth retrieval at the slope scale is confirmed by the validation in Davos, where the HS mapped with our model was compared with measurements derived from a photogrammetric survey. Indeed, the RMSE and the MAE were even lower than the ones related to the Tromsø validation area. From Fig. 1(a), it is evident that the largest overestimations are concentrated in the upper part of the mountain slope, where the steepest gradients occur and the snowpack may have experienced variations (e.g., snow creep) during the three days between the Sentinel-1 GRD acquisitions and the UAV photogrammetric survey. On the other hand, the largest underestimations are found in the lower part of the slope, where the snowpack is very thin.*

*5.4 Model limitations*

*Finally, when mapping HS over the validation area near Davos, we observed the presence of few patchy outliers. These anomalous pixels likely resulted from the sensitivity of the $SIsar$ index to local variations in snow cover conditions*

*or changes in soil properties between the snow-covered and summer reference acquisitions. However, these outliers could be effectively reduced by applying a median filter over a small pixel window (see Section 3.2). We therefore recommend using this filter as a post-processing step.*

On behalf of all the authors,

Alberto Mariani

---

## Author Comment (AC2)

Reply to Referee 2

This paper presents a novel approach to high-resolution snow depth monitoring by introducing the Snow Index SAR (SIsar), derived from dual-polarimetric Sentinel-1 data. The SIsar is defined as the difference between the Dual Polarimetric Radar Vegetation Index (DpRVIc) computed under snow-covered and average snow-free conditions. The study effectively demonstrates a correlation between SIsar and key snowpack variables, notably snowpack height and snow water equivalent. A critical finding and significant contribution of this work is the identification and subsequent correction for the influence of the LIA on the SIsar index, which improves snow depth estimations.

While the results are indeed promising and the methodology for LIA compensation shows great potential, the work, while novel in its application to Sentinel-1, resembles a substantial body of research conducted in the 1990s on C-band radar signatures of snow. Specifically, the seminal works by Kendra, Sarabandi, Ulaby, Strozzi, Wiesmann, and Mätzler (among the others) are directly relevant. These earlier studies explored C-band data collection under varying incidence angles and snow conditions, alongside the rigorous theoretical modeling of scattering mechanisms. The absence of these foundational references, and a more comprehensive literature review in general, is a significant oversight and detracts from the paper academic rigor. I strongly recommend the authors consult the (extensive) literature to provide proper context and build upon established and robust knowledge. It is highly recommended that the authors are revisiting and applying similar state of the art investigative principles for the most contemporary Sentinel-1 data. This would provide a much stronger theoretical foundation for the proposed SIsar index.

A fundamental question arises regarding the underlying electromagnetic justification of the proposed SIsar index. The paper suggests that SIsar is sensitive to snowpack properties, implying that the DpRVIc index effectively discriminates between volume and surface scattering through its relationship with VV and VH intensities and its capacity to describe depolarization (L68). However, this assertion requires further verification. Volume scattering is not the sole mechanism for depolarization; for instance, double-bounce effects, prevalent in very rough alpine terrain and due to strong individual scatterers, also may significantly depolarize the radar signal. The rationale for subtracting the summer mean DpRVIc from the snow-covered DpRVIc needs clearer justification. In alpine environments, surface scattering is a dominant factor, and its variability during summer, driven by soil moisture fluctuations, contrasts with its near-constant state in potentially frozen, high-altitude terrain during snow accumulation. Given this, it is difficult to see how this subtraction effectively isolates the scattering attributable solely to the snowpack. In this context, it is challenging to conceptualize how the specific proposed relationship between polarimetric intensities effectively discriminates between the different scattering mechanisms (surface and volume), particularly considering the observation (as highlighted in the response to Reviewer 1) that snow-free VV (across all LIAs) and VH (for LIAs from 0-25 and 70-90) backscattering are consistently higher than snow-covered backscattering, similar to findings by Strozzi et al. (1997). This indicates that the influence of snow volume scattering might be obscured by the prevailing ground contribution, consequently hindering the ability to derive meaningful insights into snow properties.

Despite these critical conceptual questions surrounding the method construction, the presented results are fairly impressive, much like those of the original Lievens et al. (2019) and following algorithms. To foster transparency and facilitate community understanding, I strongly recommend that the authors present the individual behaviors of the VV and VH signals in their paper (mean and std if more than one pixel is used and example also with only one pixel to see the impact of the speckle noise), allowing for direct comparison with the findings of Kendra et al 1998 and Strozzi et al, 1997, alongside the behavior of the DpRVIc index. Following the rationale and addressing the primary doubts raised in both Strozzi et al. (1997) and Kendra et al. (1998), I suggest that the calibration test sites should be as homogeneous as possible and thoroughly characterized. This means ensuring: uniform land cover, minimal presence of large scatterers (e.g., large rocks or boulders), absence of vegetation, consistent aspect angles, and comprehensive insights into snow conditions. If diverse "object classes" are necessary for the study, they should be clearly defined and analyzed separately, with distinct plots presented for each, similar to the approach adopted by Strozzi et al. (1997). This will provide clear evidence that varying snow depths produce significantly different backscattering responses in SIsar (even if the surface scattering change). Moreover, adopting such rigorous site selection and characterization will significantly enhance the scientific value of your work, providing critical elements for future research even in the absence of a definitive electromagnetic explanation.

Dear Carlo Marin

Thank you for your detailed and constructive comments on our manuscript, especially for the valuable literature recommendations. We believe your suggestions will significantly improve the quality of the manuscript and, in particular, of the introduction. We start our reply with some general comments.

Based on the theory and our observations we agree on the fact that the $DpRVI_c$ index alone cannot discriminate the snow backscatter contribution, since depolarization occur also due to other mechanisms. However, the snow volumetric backscatter can be significantly isolated if you work in change detection, so detracting the depolarization contribution of soil macroscopic properties characteristics (i.e., the land cover, such the presence of large boulders or shrubs). As better detailed in the reply to Eq 1 and Eq 2, by normalizing each scene with a summer reference, representative of the average $DpRVI_c$ index values due to land cover, we can reasonably assume that the effect of soil and vegetation cover is largely negligible. Note that the work of Feng et al. (2024) follows a similar idea. Regarding the working principle of $SIsar$ and $DpRVI_c$ indices for snow depth ($HS$) retrieval, we have also provided explanations in the responses. However, we would like to highlight a few additional points. Firstly, we demonstrated that gammaVH is slightly higher for certain $LIA$ angles in the presence of snow than in summer (see Figure 2 below): these specific $LIA$ values are precisely those for which we found $SIsar$ suitable for $HS$ retrieval. Secondly, we refer to the recent work by Brangers et al. (2024), where a tower-based C-band radar system was used in conjunction with Sentinel-1 and snow stratigraphy observations: that study demonstrates not only that snow depth retrieval using C-band SAR can perform well, but also that:

- A volumetric backscatter contribution is present in the snowpack and is reflected by an increase in gammaVH, whereas gammaVV is generally not sensitive to increases in snow depth;

- This volumetric dry snow backscatter can, in some cases, be of the same magnitude as the ground contribution;

- The use of VH/VV ratios or derived indices (such as $DpRVI_c$) is a sound approach, as it can help mitigate the influence of varying snowpack properties, and increase the sensitiveness of the VH band alone to the snow depth.

These results align with our findings, especially when looking at the trends in VV, VH, $DpRVI_c$, and ultimately $SIsar$ with $LIA$ and $HS$, as presented in the manuscript and in our response to Reviewer 1. Furthermore, after reviewing the work of Strozzi et al. (1997), we can show that our results are consistent with their findings. To support this, we include a comparison using VV, VH, $DpRVI_c$, and $SIsar$ values directly derived from the data presented in that paper (see Figure 1, to be compared with Figure 2 below). Please note that in this validation the $SIsar$ index is approximated from the available data in Strozzi et al. (1997) where only one set of backscatter coefficient observations for varying $LIA$ values was available for summer, which was used as the summer reference. Nonetheless, the trends observed for the $SIsar$ index in our study closely resembles the one derived from Strozzi's data. Based on these new observations, we can confidently propose the following interpretation of the working principle:

- gammaVV is not sensitive to snow depth, except at low $LIA$ values, as also demonstrated in previously cited works;

- gammaVH is sensitive to snow depth and may even exceed summer values for certain $LIA$ ranges (see Figure 2 below);

- the variations of such quantities are limited and therefore they are not suitable for $HS$ retrieval alone. For this specific reason we believe that Strozzi concluded that at C-band is impossible to sense variation in snowpack depth.

However, combining gammaVV and gammaVH in the $DpRVI_c$ index potentially allow to estimate $HS$, indeed for sufficiently high $LIAs$ its values in summer are consistently lower than those observed under snow-covered conditions, both in our data and in Strozzi's measurements. Based on this hipothesis we based our study and we reserved the demonstration of this with our results. The only exception, as explained, is for $LIA < 30°$, where this fact is not true and the $SIsar$ index becomes negative, limiting its reliability for $HS$ retrieval. In general, we will be more caution in the final version of the manuscript with the explanations of the backscatter behaviour, which as suggested would require more detailed and specific experiments. As a consequence we will modify and clarify these reasonament within the introduction and the discussion.

Concerning, the calibration sites, see reply to L235 and, in particular, reply to referee 1.

To conclude, we would like to reiterate that the present study is focused on highlighting and verifying the existence of a dependence between the $SIsar$ index (or $DpRVI_c$) and the $LIA$, and on proposing a method to compensate for it within an algorithmic framework, with the final aim of improve the spatial resolution of snow depth retrieval algorithm. We acknowledge that the backscattered mechanism in dry snow is complex and still poorly understood, and further studies—supported by specific and precise ground-truth data on snowpack properties across varying $LIAs$, snowpack and ground conditions—are necessary to

[Figure]

Figure 1: Revised results from Strozzi et al. (1997).

refine the relationship between $DpRVI_c$, SAR backscatter, and $SIsar$ with $LIA$ and snowpack properties.

Finally, please find below our replies (in blue) describing how we will address your comments in the revised manuscript.

**Detail comments:**
The current title seems a bit too generic. I suggest incorporating your finding of a quadratic relationship between the SIsar index and LIA directly into the title (specifying what a mathematical model is). This would immediately highlight a significant contribution of your work.

We will think about this at the end of the peer review process. The problem of talking about the quadratic relationship in the title is that it requires the knowledge of the $SIsar$ index, which is introduced in our paper.

L35: From a scientific standpoint, the initial question revolved around whether microwave can effectively be employed to extract pertinent information from snow. The selection of SAR in this context is primarily driven by its inherent capability to provide the requisite spatial resolution for detailed analysis especially in mountain areas.

We will explain better the "history" of snow remote sensing in this part of the Introduction, highlighting better the potential of SAR. We will also cite additional papers, as you previously suggested.

L38: for SAR signals interactions with snow please read the fundamentals literature and books from Ulaby, Mätzler, Picard, Löwe, Tsang and many others. The dielectric constant, which is the real part of the permittivity is only one small ingredient. I suggest to read the review from Mätzler "Applications of the interaction of microwaves with the natural snow cover" (written in 1989!) to better shape the introduction.

We will read the suggested books and articles in order to improve the introduction. We will also cite some of such papers in the revised version of the manuscript.

L47: this seems not be true. See Strozzi et al, 1997.

In the conclusions of Strozzi et al. (1997) they found that that the backscatter at Ku-band increases more than the backscatter at C-band in presence of a snow cover. In particular, it is reported that *the increase in backscatter is pronounced only at ku-band.* The sentence at L47 comes directly from the conclusions of the cited paper (Tsang et.al. 2022), which agree with the findings of Strozzi et al. (1997). However, none of the mentioned work clearly talk about volumetric backscatter, so we will modify the sentence talking in general about backscatter and not volumetric backscatter (we will also check the reminder of the paper to be sure that this refuse will not be present in the revised version of the manuscript).

L68: While I am not a polarimetry expert, I question whether volume scattering is the exclusive source of depolarization, as individual scatterers or very rough surfaces can also induce polarization rotation.

We fully agree with this observation; however, it is important to note that, both in our study and in the work by Feng et al. (2024), where the $DpRVI_c$ index was shown to outperform other indices in snow depth ($HS$) retrieval, such index is not used in its absolute form, but always in a relative manner. Specifically, in our case, and precisely for this reason, we subtract the average $DpRVI_c$ index value observed under snow-free conditions from the $DpRVI_c$ index value of the winter scene. This approach allows us to effectively remove any depolarization contribution from the underlying soil, under the assumption that its macroscopic properties do not change significantly between the summer and winter acquisitions. In particular, the surface roughness of the soil is assumed to remain constant, considering the short temporal interval between the summer reference acquisition and the corresponding winter scenes.

Study area: The land cover and soil type require characterization. Which DEM was utilized?

We will provide more details in the revised version of the manuscript. We will add some details related to the characteristics of the soil when buried under snow, which varied over the seasons.

Snowpack modeling: To ensure clarity and adhere to established standards, I suggest adopting the symbols from Appendix D of The International Classification for Seasonal Snow on the Ground. For instance, E is typically used for grain size, and $\rho$ for the density. Presently, $\rho HS$ might be confused with SWE.

We will make these changes in the revised version of the manuscript.

L166: Regarding the measurement, could you specify whether the units are in percent by volume or millimeters? It is also worth considering that averaging across the entire snowpack might introduce inaccuracies, given that microwave signal attenuation, particularly in the presence of significant wet superficial layer that can limit the penetration depth.

We will specify the unit of measure in the revised version of the manuscript. We are aware that averaging trough the entire snowpack is a limitation, however this choice is due to the fact that we used snow properties simulated with the snow cover model SNOWPACK. Indeed, finding an index suitable for our random forest and statistical analysis which considers both the $LWC$ and its position within the snowpack is not possible. Anyway, the statistical analysis is solely intended to identify the snowpack variable most strongly correlated with the $SIsar$ index, not to verify the precise signature of each parameter to the backscatter. Moreover, we highlight that the results of the statistical analysis are in line with the findings of other cited works (as reported in our manuscript). Concerning the presence of wet layers, we analyzed their effects in Appendix A.

L170: did you use the "projected LIA" from SNAP?

We used the local incidence angle ($LIA$), not the projected local incidence

angle ($PLIA$) from SNAP. The backscatter strongly depends on the true orientation of the surface relative to the radar beam, especially in non-flat terrains like snow-covered slopes. The $LIA$ accounts for the local surface normal, incorporating slope and relief, and is therefore more physically consistent with the radar–surface interaction. Snow is a highly anisotropic medium in terms of radiometric behavior: even small changes in local incidence angle can lead to significant differences in the radar return. For this reason, we opted for the $LIA$, which better reflects the physical scattering conditions at the surface. Finally note that the $LIA$ has beed used in other works for snow avalanches mapping (Tompkin and Leinss, 2021; Keskinen et al., 2022).

L174: Could you please explain the rationale for using the refined Lee filter?

Firstly, we got inspired by the majority of work in which preprocessing of SAR data for snow or avalanche detection purpose is presented. It would be very interesting to do a sensitive analysis changing the filtering technique in the preprocessing phase, however we believe that this is out of scope for our manuscript. We also had a closer look to temporal filtering techniques, however we decided to not use this for mainly two reasons:

1. Our work aims to improve the $HS$ detection for operative scopes, like the support of avalanche forecasting activities. Temporal filter requires a certain number of acquisitions for the preprocessing and this can require a very high computational cost, especially for large areas.

2. The temporal filter requires a certain temporal radiometric stability in the scatterer, and we assume that this condition is missed in presence of dry snow.

In conclusion, in this study, the Refined Lee Filter was chosen to ensure consistent, per-image noise reduction, without introducing artifacts that could arise from temporal averaging across dynamically changing scenes.

L175: Given the S1 SAR sensor native resolution of 5m x 20m and the subsequent application of spatial filtering, a final resolution of 10m appears optimistic. A resolution closer to 20m seems more analytically consistent with these parameters.

The 10 m resolution results from the preprocessing activity performed in SNAP as described. The word "resampled" is a refuse which will be corrected in the final version of the manuscript, as described in the Reply to Referee 1. Anyway, we think that this highlights the validity of our model: even without an high resolution we were able to obtain promising results.

Eq 1 and Eq 2: can you better justify why you choose this index and why you made the difference between the indexes? What does it change if you change the reference? Would it be more appropriate to select as reference the initial backscattering after a significant accumulation, accounting for soil insulation that will persist for all the season long (if no permafrost is present)?

Eq. 1: As described in the introduction, the $DpRVI_c$ index follows directly from the work of Feng et al. (2024), who recently demonstrated that this index, which can be simply approximated from a GRD product (a great advantage from a computational point of view) outperforms all other dual-polarimetric indexes in snow depth retrieval.

Eq. 2: As already explained in a previous answer (see reply to L68), as observed by Feng et al. (2024), the $DpRVI_c$ index cannot isolate the depolarization resulting from the snowpack since there are also other sources of depolarization (land cover and ground properties). We therefore decided to

subtract to it the average $DpRVI_c$ index value obtained under snow-free condition. In this way we could assume that the $SIsar$ index variations ($SIsar = DpRVI_c - DpRVI_c^{summer\ reference}$) are only due to variations occurred between the summer reference and the subsequent winter, which are negligible at least in terms of land cover and macroscopic ground properties. Moreover, we believe that the usage of an average between many snow free images related to the closest-in-time summer reduces the effects of soil properties variations. This technique is widely used and recognized in several SAR-based snow monitoring approaches, like the famous Nagler's method for wet snow retrieval. Furthermore, the usage of an average summer reference does not require any knowledge on the temporal weather condition, which is fundamental in the poorly monitored areas. It is interesting that a relationship between the $DpRVI_c$ and $HS$ like Eq. (8) cannot hold. Indeed, as reported in the following image, that we will also add in the revised version of the manuscript, such index shows a dependency on the $LIA$ even in absence of snow (see the yellow curve in subgraph (b)). This further supports our approach.

[Figure]

Figure 2: Mean $SIsar$ (a), $DpRVI_c$ (b), $\gamma_{VH}^0$ (c), and $\gamma_{VV}^0$ (d) values, sampled from the $LIA$ sampling areas, grouped into $LIA$ classes of $2°$ width. The blue color scale represents the increasing average $HS$ values measured by the two AWSs for different acquisition dates along the 2023-2024 season. Yellow points represent the values from an acquisition date in summer 2023.

Finally, we decided to not subtract the initial backscatter after each accumulation for a purely strategic reason. As previously explained, our work is aimed at applications (avalanche forecasting for remote regions, for example). With such method, in order to derive the snow depth in a certain date within the winter season, one must analyze several seasons, significantly increasing the computational and storage costs. Thus we decided to use this simplified method and we showed the benefits. The only other way to avoid the preprocessing of an entire time series is using the average $DpRVI_c$ index value of the last images before the snow accumulation start in autumn/early winter. This would produce a reference more representative of the soil conditions below the upcoming snowpack, but again it requires a continuous knowledge of the snow cover (which is not possible to have for remote regions). We will explain this in the discussion of the revised version of the manuscript.

It would be beneficial to understand why the aspect angle i.e., the angle between the sensor flight direction and the surfaces orientation, was not considered into the analysis. This raises the question of whether the surface and snow are being implicitly assumed as isotropic mediums

As explained in the Introduction (L85), our initial objective was trying to understand what happens, at a small-scale (e.g., single mountain slope), when the SAR signal penetrates more or less snow (i.e., the $LIA$ varies). We acknowledge the importance of the aspect angle in radar backscatter, however, in our study, our aim is to use SAR observations to detect and characterize variations in the snowpack properties themselves, without introducing assumptions about their directional dependence. Including aspect angle in the analysis would imply incorporating a prior model of directional variability (e.g., different snow or surface behavior depending on slope orientation). But in our case, we intend to observe whether such variability exists and how it manifests in the SAR response. In other words, considering aspect would mean assuming that snow characteristics change with orientation, but our goal is to assess whether and how they change, not to impose it beforehand. We therefore opted to analyze the data using only the $LIA$, which accounts for slope steepness but not orientation.

L204: It is important to carefully review the symbols used here to ensure they are consistent across the entire paper.

We did this and we ensure you that there are no errors.

L206: I am seeking further clarification regarding the methodology. Specifically, the justification for integrating the Random Forest algorithm to find the most predominant features, with a Least Squares method later, needs more comprehensive explanation (at least to me in the present form). Can you do the same using only one method?

In order to understand which snowpack variables have a strong relationship with our $SIsar$ index we used two different approaches. The random forest was used to obtain an initial idea of the presence or the absence of such relationship. We say "idea" because the sample size is not very large. We therefore performed a statistical analysis to study such relationships with another mathematical technique. The results of the two analysis are in line with each other (as observed in our paper) and they strengthen each other. However, we explain that the second analysis is the most important one for two reasons: firstly, through several statistical tests (for example, the $p$-value) we were able to prove that those results are reliable; secondly, through other tests (for example, the Pearson's correlation test) we were able to have an idea of the kind of relationship. At the end, we used the least squares method to derive the precise relationship. Obviously, we could directly do a linear or quadratic regression but in that case the results would not have been validated with a statistical analysis.

L235: It is important to note that snow distribution is highly preferential, influenced by both topography and meteorological conditions. This inherent variability is precisely why utilizing such high-resolution data is crucial for accurate analysis.

This is a limitation that also Referee 1 was concerned about (see also that reply). In general, we explained that the reliability of those measurements is demonstrated by the fact that the avalanche forecasters of the Livigno Avalanche Center carry out weekly measurements in different places of the valley and that similar data were already used for different papers during the past years, see for example: (Monti et al., 2014), (Monti et al., 2016), *Simulation of snow management in Alpine ski resorts using three different snow models* (Hanzer et

al., *Cold Regions Science and Technology*, 172, 102995, 2020). Moreover, we decided to carry out an additional validation with photogrammetric data (see again reply to referee 1) to validate our model with a large dataset: remarkably, with this additional validation we obtained errors even lower than with the other validations, further supporting the validity of the calibration of our model.

L297: just shadow.

We will change the wording as suggested in the revised version of the manuscript.

Fig 2. Following the criteria outlined by Nagler and Rott 2000, the presence of layover is identified for LIA less than 17° and shadow regions for LIA greater than 78°. Ensuring consistency with these thresholds throughout the analysis is appreciated for better reading the plots. Additionally, please standardize the LIA classification scheme across all plots for coherence. Could you also provide comments on the source of the standard deviation and outliers, as raised in the general comments? Finally, it would be beneficial to discuss the influence of aspect angle and the different land cover types on the results.

We will add the following sentences in the revised version of the manuscript (in Section 5.2, Line 353). *As shown in Fig. 2, the presence of outliers in the $SIsar$ index values across the $LIA$ classes is notable. However, such variability is common in this type of experiment and can be attributed to various sources of error, including wind redistribution effects, speckle noise, and spatial variability in snowpack properties.* Concerning the $LIA$ classes, we chose classes of 5° width for the box plot just to improve graph visualization since a 2° width would lead to a graph difficult to read. However, for the mathematical analysis and the subsequent graphs we chose 2° classes to improve the accuracy of the model. We discussed in the paper (L299) why it is not possible to consider also $LIAs$ lower than 30° in the implementation of the final model. This fact is also further supported by the behaviors of $\gamma_{VV}$ and $\gamma_{VH}$ (recall Fig. 2 above), as already discussed in the reply to referee 1 (see reply to Line 366). Finally, for the aspect angle see a previous reply.

Fig3. Could you please explain why, beyond an incidence angle of 60 degrees, the backscattering for a snow depth of 60 cm becomes higher in terms of SIsar?

Unfortunately, a clear explanation for this behavior is not readily available. One hypothesis is that, on 12 November 2024, the basal layer of the snowpack was generally wet, as indicated by both simulations and manual observations. As reported in Appendix A, such conditions can lead to overestimations. However, this wet basal layer cannot be assumed to be homogeneous across the entire study area. Therefore, we cannot exclude the possibility of preferential patterns in this snowpack property with respect to $LIA$ on this date. In this analysis, we considered these values as outliers. We acknowledge that further studies, supported by specific ground-truth data on snowpack properties across varying $LIA$ conditions, are needed to better refine the relationship between the $DpRVIc$ index, SAR backscatter, and the $SIsar$ index as a function of the $LIA$. However, this lies beyond the scope of the present study, which aims to highlight and verify the existence of a dependence, and also present a way to compensate it in an algorithm.

Regarding Figures 5 and 6, were these HS measurements obtained manually?

As explained in the methods section, these $HS$ values are obtained manually for the Tromso validation area (Figure 6), while they derive from automatic weather stations measurements for the Livigno area (Figure 5).

L321: Could you elaborate on the statement that the snowpack contains a larger ice component (if I understood what a "snow mass inside a snowpack" is)? Specifically, how was it determined that this implies a higher concentration of grains and discontinuities? Furthermore, I am not entirely convinced that

snow layers with varying relative permittivity alone can depolarize the signal; this would be an interesting hypothesis to demonstrate.

*At L321 we have taken up an idea already presented in the work of Lievens et al. (2019), cited in the subsequent sentence, where it is suggested the V band signal depolarization increase with increasing discontinuities within the snowpack. A similar behavior was confirmed in the work of Keskinen et al. (2022) on avalanche deposits detection, we will add this citation at the end of the sentence. However, as suggested by Referee 1, we will change the wording in the following way: One possible explanation for this increase is that thicker and denser snowpacks contain larger snow mass and more grains per unit volume of snow. The fact that snow layers with varying relative permittivity alone can depolarize the signal is not obvious and out of the scope of our paper, indeed, as explained in the manuscript, this is just a possible explanation. Finally, the fact that thicker and denser snowpacks contain a larger snow mass, more grains, and more potential discontinuities follows from the fact that they have a large number of different layers, each characterized by different physical properties.*

L322 and on. For a more pertinent comparison, I suggest referencing Kendra et al. (1998) and Strozzi et al. (1997), given their direct relevance, as opposed to broader X-band research.

*We will do so in the revised version of the manuscript.*

L355: Given the likely significant disparity in scatterer size between vegetation and snow, I am not convinced that a direct or simple analogy between their scattering properties is entirely straightforward

*This analogy was proposed by Feng et al. (2024), we will report a citation at the end of the sentence in the revised version of the manuscript.*

Table 2: Could you please specify the total number of validation points

*We will add a column with that number in the revised version of the manuscript.*

Fig 7: I am not certain I have fully understood the representation in this figure. It does not appear to align with the solutions of any radiative transfer equations commonly used for the snow problem that I am familiar with. Specifically, could you explain why the VV polarization appears unaffected by the presence of snow? Furthermore, the contribution of surface scattering, which could be dominant, seems to be unrepresented. Please refer to Figure 6 of the paper of Kendra et al, 1998 for a complete first order volume scattering mechanism representation for a snow layer.

*As explained in the reply to Referee 1, we will remove this picture from the final version of our manuscript. See that reply for details.*

L401: If the geographical aspect varies E-W (or W-E) while the slope remains constant, the LIA should remain the same.

*Absolutely. With that sentence, our intention was simply to highlight that the different field measurements correspond to different $LIAs$ because they were collected on various mountain slopes, which are naturally not homogeneous. We did not mean to say that a change in aspect implies a change in the $LIA$; rather, our point is that when the slope itself changes, the associated $LIAs$ almost certainly differ as well, since the slopes are not identical. In any case, we will revise this sentence in the updated manuscript to clarify that the field measurements were taken from different mountain slopes and therefore span a wide range of $LIAs$.*

**References:**

J. R. Kendra, K. Sarabandi and F. T. Ulaby, "Radar measurements of snow: experiment and analysis," in IEEE Transactions on Geoscience and Remote Sensing, vol. 36, no. 3, pp. 864-879, May 1998, doi: 10.1109/36.673679.

T. Strozzi, A. Wiesmann and C. Mätzler, "Active microwave signatures of snow covers at 5.3 and 35 GHz," in Radio Science, vol. 32, no. 2, pp. 479-495, March-April 1997, doi: 10.1029/96RS03777

C. Tompkin and S. Leinss, "Backscatter Characteristics of Snow Avalanches for Mapping With Local Resolution Weighting," in IEEE Journal of Selected Topics in Applied Earth Observations and Remote Sensing, vol. 14, pp. 4452-4464, 2021, doi: 10.1109/JSTARS.2021.3074418

Zachary Keskinen, Jordy Hendrikx, Markus Eckerstorfer, and Karl Birkeland, "Satellite detection of snow avalanches using Sentinel-1 in a transitional snow climate", Cold Regions Science and Technology, vol. 199, 2022, 103558, https://doi.org/10.1016/j.coldregions.2022.103558

On behalf of all the authors,

Alberto Mariani